# Costimulation of type-2 innate lymphoid cells by GITR promotes effector function and ameliorates type 2 diabetes

Lauriane Galle-Treger[1], Ishwarya Sankaranarayanan[1], Benjamin P. Hurrell[1], Emily Howard[1], Richard Lo[1], Hadi Maazi [1], Gavin Lewis[2], Homayon Banie[2], Alan L. Epstein[3], Peisheng Hu[3], Virender K. Rehan[4], Frank D. Gilliland[5], Hooman Allayee[6], Pejman Soroosh[2], Arlene H. Sharpe[7] & Omid Akbari[1]

Metabolic syndrome is characterized by disturbances in glucose homeostasis and the development of low-grade systemic inflammation, which increase the risk to develop type 2 diabetes mellitus (T2DM). Type-2 innate lymphoid cells (ILC2s) are a recently discovered immune population secreting Th2 cytokines. While previous studies show how ILC2s can play a critical role in the regulation of metabolic homeostasis in the adipose tissue, a therapeutic target capable of modulating ILC2 activation has yet to be identified. Here, we show that GITR, a member of the TNF superfamily, is expressed on both murine and human ILC2s. Strikingly, we demonstrate that GITR engagement of activated, but not naïve, ILC2s improves glucose homeostasis, resulting in both protection against insulin resistance onset and amelioration of established insulin- resistance. Together, these results highlight the critical role of GITR as a novel therapeutic molecule against T2DM and its fundamental role as an immune checkpoint for activated ILC2s.

[1] Department of Molecular Microbiology and Immunology, Keck School of Medicine, University of Southern California, 1450 Biggy St NRT 5509, Los Angeles, CA 90033, USA. [2] Janssen Research and Development, 3210 Merryfield Row, San Diego, CA 92121, USA. [3] Department of Pathology, Keck School of Medicine, University of Southern California, 2011 Zonal Ave HMR, Los Angeles, CA 90033, USA. [4] Division of Neonatology, Harbor-UCLA Medical Center, 1000W Carson St, Torrance, CA 90502, USA. [5] Department of Preventive Medicine, Division of Environmental Health, University of Southern California, 2001 N Soto St SSB 230G, Los Angeles, CA 90033, USA. [6] Departments of Preventive Medicine and Biochemistry & Molecular Medicine, Keck School of Medicine, University of Southern California, 2250 Alcazar St CSC 2202, Los Angeles, CA 90033, USA. [7] Department of Microbiology and Immunobiology, Harvard Medical School, 77 Ave Louis Pasteur, Boston, MA 02115, USA. Correspondence and requests for materials should be addressed to O.A. (email: akbari@usc.edu)

Chronic obesity is associated with the development of low-grade systemic inflammation and recruitment of inflammatory immune cells to metabolic active tissues, such as the adipose tissue, liver, and muscle. This pro-inflammatory environment promotes insulin resistance and elevation of blood glucose levels, predisposing patients to development of T2DM[1,2]. The metabolic disturbances associated with obesity-induced inflammation in the adipose tissue are thought to involve the expansion and a remodeling of immune cells, particularly in visceral adipose tissue (VAT)[3]. In this regard, type 2 innate lymphoid cells (ILC2s) are among the various immune cells that have recently been identified to be present in the VAT[4,5]. Similar to Th2 cells, activated ILC2s can produce significant amounts of IL-5 and IL-13, and can therefore play important roles in regulating metabolic homeostasis in VAT[6–9]. For example, the production of IL-13 induces differentiation of macrophages towards an anti-inflammatory phenotype, referred to as alternatively activated macrophages (AAMs), whereas IL-5 plays a crucial role in the activation and recruitment of eosinophils, which in turn secrete most of the IL-4 required for the maintenance of AAMs[4,10]. In addition, ILC2s express MHC class II and costimulatory molecules such as CD80, CD86[11], ICOS[12,13], and we further showed that ILC2s also express ICOS-L[14].

Glucocorticoid-induced tumor necrosis factor receptor (GITR), also known as TNFRSF18, is a member of the TNFR superfamily which is expressed on multiple cell types, including on CD4+ and CD8+ T lymphocytes[15,16]. GITR is currently of interest to immunologists as a co-stimulatory immune checkpoint molecule[17,18]. It is upregulated in the context of inflammation and acts as an important costimulatory signal in T lymphocyte subpopulations, as studies have shown that engagement of GITR with its ligand (GITRL) in vivo induces T cell expansion and cytokine production[19–23]. Moreover, GITR was also described as a marker for Treg activation in animal models as its engagement on Tregs led to expansion and paralleled loss of suppressor activity in vitro[15,24].

In this study, we evaluate the effects of GITR engagement on ILC2s in the context of insulin resistance. We find, that human and mouse ILC2s from the VAT express the GITR costimulatory receptor. Using a GITR specific agonist and a mouse model of GITR deficiency, we discover that GITR engagement not only protects against the development of T2DM onset but can also ameliorate established T2DM. We further demonstrate that the protective role of the GITR agonist is IL-13 dependent and that engagement of GITR induces activated ILC2 effector function while increasing the expression of the critical inflammatory modulator NF-κB. Furthermore, GITR is expressed on both naïve and activated ILC2s, highlighting the role of GITR as an immune checkpoint molecule capable of exclusively costimulating activated ILC2s. Our findings provide new insights on GITR's role in ILC2s and introduce GITR engagement as an ideal therapeutic target against T2DM.

## Results

**GITR is expressed on ILC2s and induces Th2 cytokines.** Based on our recent observations that ICOS, a costimulatory molecule expressed by immune cells, was expressed by ILC2s and modulated their effector function and homeostasis[13], we first assessed whether GITR was also expressed on naïve and IL-33-activated ILC2s. Mice received intraperitoneal IL-33 or PBS on 3 consecutive days. On day four, ILC2s from visceral adipose tissue (VAT) were analyzed by flow cytometry and gated as lineage⁻CD45+ IL-7R+ and ST2+ (Fig. 1a). Further analysis of the cell-surface phenotype of ILC2s showed that both naïve and IL-33-induced ILC2s had high expression of GITR, although there were no differences in expression between naïve and activated cells (Fig. 1b). To assess the effect of GITR engagement on ILC2 activation, we measured the levels of cytokine secretion in presence of specific GITR agonist, DTA-1 or the isotype control. Freshly isolated VAT-derived ILC2s were first stimulated in vitro for 48 h with specific GITR agonist DTA-1 or the isotype control (naïve ILC2s, Fig. 1c). In parallel, to measure the effect of GITR in

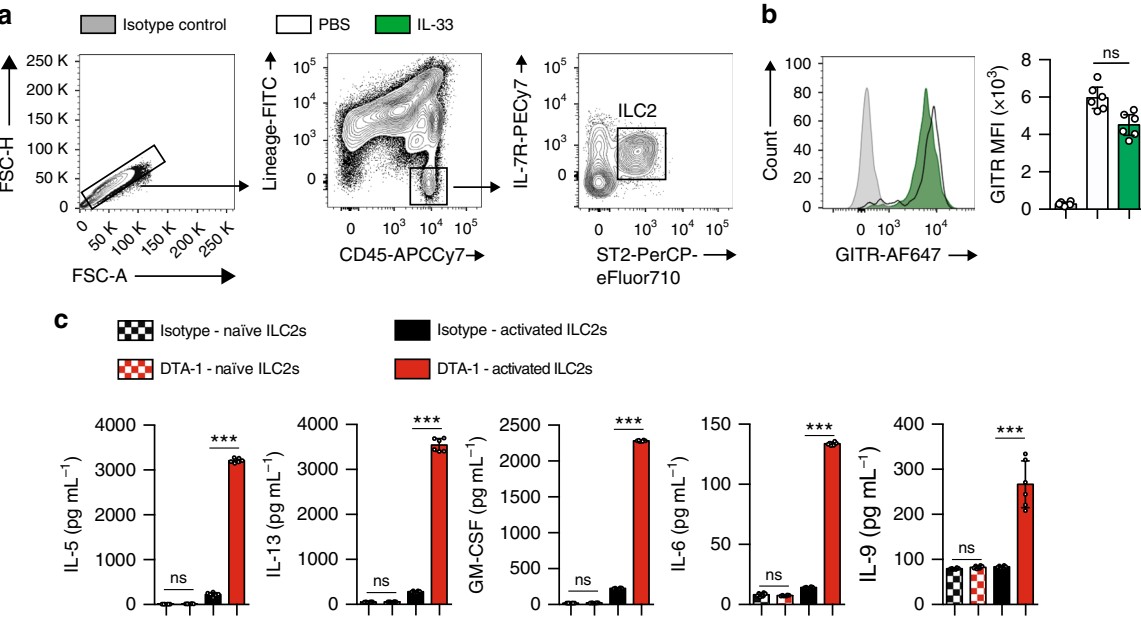

**Fig. 1** GITR is expressed on ILC2s and induces Th2 cytokine secretion in activated ILC2s. **a** Gating strategy of Lin⁻CD45+IL-7R+ST2+ ILC2 cells. **b** Expression of GITR in naïve (white) and IL-33-activated (green) murine white adipose tissue-derived ILC2s compared to the isotype control (gray). Corresponding quantitation of GITR expression shown as MFI+/− SEM, n = 6. **c** Visceral adipose tissue (VAT) resident ILC2s were isolated from a cohort of naïve C57BL/6 mice and a cohort of activated C57BL/6 mice intraperitoneally challenged with rmIL-33 on 3 consecutive days. Naïve and activated VAT ILC2s were cultured with recombinant mouse (rm) IL-2 and rmIL-7 with DTA-1 (5 μg/mL) or isotype control for 48 h. The levels of IL-5, IL-13, GM-CSF, IL-6, and IL-9 were measured by Luminex on the culture supernatants, n = 6. Error bars are the mean ± SEM. Student's t-test, ***p < 0.001, ns, non-significant

an activated context, freshly isolated VAT-derived ILC2s were similarly stimulated in presence of recombinant mouse (rm)IL-33 for 48 h with DTA-1 or the isotype control (activated ILC2s, Fig. 1c). Following incubation, cytokine secretion was then measured on the cell culture supernatants by Luminex. Cytokine production by naïve ILC2s was not affected by DTA-1 treatment (Fig. 1c). In contrast, when activated with rmIL-33, GITR engagement induced secretion of high amounts of IL-5, IL-13, GM-CSF, IL-6, and IL-9 compared to controls (Fig. 1c). Taken together, these results show that even though GITR is expressed on both naïve and activated ILC2s, the induction of Th2 cytokine secretion after GITR engagement requires ILC2s to be activated, suggesting a co-stimulatory role of the GITR receptor in ILC2s.

**Engagement of GITR protects against metabolic disturbances.** It has been previously reported that activation of ILC2s in VAT limits adiposity and insulin resistance in mice fed a high fat diet (HFD)[4,6]. We therefore next examined whether activation of ILC2s through GITR engagement could similarly prevent the development of insulin resistance in vivo using C57BL/6 mice fed either a normal chow diet (NCD) or a HFD to induce obesity. We treated a cohort of C57BL/6 mice fed either a NCD or a HFD with intraperitoneal injections of DTA-1 (1 mg/mouse) or the isotype control every four days for 14 weeks (Fig. 2a), and assessed a variety of metabolic parameters. Both DTA-1 and isotype control treated mice fed a HFD had similar increased weight gain compared to NCD fed mice and developed diet-induce obesity (Fig. 2b). Despite the fact that we observed no difference in total weight gain between groups on HFD, GITR engagement resulted in a decrease of VAT weight (Supplementary Figure 1a). Strikingly, DTA-1 treated mice fed a HFD had lower fasting blood glucose levels as compared to isotype-treated mice fed a HFD (Fig. 2c). Additionally, DTA-1 treated mice fed a HFD showed improvements in glucose tolerance and insulin sensitivity compared to the isotype control mice as shown by intraperitoneal glucose tolerance tests (ip-GTTs) and insulin tolerance tests (ITTs), respectively (Fig. 2d, e). We next measured ILC2 numbers and effector functions of VAT resident ILC2s in vivo in response to GITR engagement. Mice fed HFD showed increased numbers of VAT resident ILC2s, associated with increased expression of IL-5 and IL-13 (Supplementary Figure 1b-d). GITR engagement in mice fed NCD showed no effect on blood glucose levels, glucose tolerance, insulin sensitivity, VAT weight and VAT ILC2 response (Fig. 2 and Supplementary Figure 1). Furthermore, we investigated the source of IL-33 and IL-25 responsible for ILC2 activation in the context of a chronic low-grade inflammation such as diet-induced obesity and metabolic syndrome. Interestingly, we observed that *IL-33* and *IL-25* were both significantly increased over time by quantitative real time PCR in mice fed a HFD compared to mice fed a NCD in VAT lysates, suggesting local secretion of IL-33 and IL-25 (Fig. 2f, g). Our results were confirmed using the mouse model of leptin deficiency Ob/Ob mice fed a regular chow diet, who spontaneously develop severe obesity associated with insulin resistance[25] (Supplementary Figure 2). Using this model, DTA-1 treatment over the course of 14 weeks also strikingly lowered fasting blood glucose levels, improved glucose tolerance as well as insulin sensitivity, altogether without affecting weight gain (Supplementary Figure 2a-e). Interestingly, treatment of mice with DTA-1 resulted in increased numbers of ILC2s in both the VAT and BM over time, as compared to control isotype control-treated mice (Supplementary Figure 2f-i). Collectively, these observations demonstrate that GITR engagement can limit the onset of obesity and improve glucose homeostasis in the context of metabolic syndrome.

**GITR engagement in ILC2s prevents type 2 diabetes.** GITR expression is not restricted to ILC2s and is also present on other immune cells such as T cells[23]. We therefore assessed if GITR engagement is effective in *Rag2* deficient mice, which lack the adaptive branch of the immune system but still have ILC2s. Therefore, a cohort of *Rag2-/-* mice were fed a HFD and either treated with DTA-1 (1 mg/mouse) or the isotype control by intraperitoneal injections every four days for 14 weeks (Fig. 3a). Similar to our previous results (Fig. 2), DTA-1 treatment did not affect weight gain compared to isotype control-treated mice (Fig. 3b). DTA-1 treatment was however associated with reduced fasting glucose and insulin concentrations in plasma (Fig. 3c, d), as well as increased glucose tolerance and insulin sensitivity during ip-GTTs and ITTs, respectively (Fig. 3e, f). To further dissect the effects of DTA-1 treatment, we analyzed VAT structure by histology (Fig. 3g). In response to DTA-1, we measured a lower leukocyte infiltration (Fig. 3g) in addition to a decrease in adipocyte size (Fig. 3h) compared with the isotype control group. Altogether, these data strongly suggest that the protective effects of GITR engagement on glucose homeostasis are independent from the adaptive immune system. To further understand the mechanisms by which GITR regulates adiposity, we phenotyped DTA-1 and isotype control treated *Rag2-/-* mice with metabolic cages and analyzed their body composition by nuclear magnetic resonance spectroscopy (NMR). Although we did not observe differences in food and water intake nor in physical activity (Supplementary Figure 3a-c), GITR engagement resulted in a decrease of fat mass percentage (Fig. 3i) and of total VAT weight (Supplementary Figure 3f). In line with this, GITR engagement induced an increase in total oxygen consumption ($VO_2$) (Fig. 3j) and energy expenditure normalized by total weight (heat) (Fig. 3k). Importantly, DTA-1-treated mice also showed increased lean mass percentage, although it was not associated with changes in liver or spleen weights (Supplementary Figure 3e and 3g-h). As expected, GITR engagement had no effect on energy expenditure when the data was normalized to lean mass (Supplementary Figure 3d). These results suggest that the metabolic improvements associated with GITR engagement could be mediated, at least in part, through increased oxidative metabolism rather than through effects on caloric intake or physical activity. As it has been previously demonstrated that ILC2 activation can induce beiging of the adipose tissue and in turn increase thermogenesis, which increases caloric expenditure[6], we assessed the expression of Uncoupling protein 1 (*Ucp1*), a critical protein involved in the thermogenic process in adipocytes. Mice treated with DTA-1 exhibited increased expression of *UCP1* expression both at the protein level by immunohistochemistry (IHC) and at the mRNA level by RT-qPCR in the VAT (Fig. 3l, m). Other genes known to be involved in browning of adipose tissue were induced by DTA-1 treatment in VAT lysates (Supplementary Figure 4). We observed higher expression of cell death activator CIDE-A (*Cidea*), PR domain zinc finger protein 16 (*Prdm16*), peroxisome proliferator-activated receptor-γ coactivator (*Pgc1a*), cytochrome oxidase subunit VIIa polypeptide (*Cox7a*), and deiodinase 2 (*Dio2*), in VAT lysates in mice treated with DTA-1 compared to isotype control group. Altogether, these results demonstrate that GITR engagement in ILC2s has protective effects on glucose homeostasis by inducing the beiging of the adipose tissue through co-stimulation of ILC2s.

**GITR agonist induces Th2 cytokines by ILC2s.** We previously observed in vitro that GITR engagement promotes IL-5 and IL-13 secretions in activated ILC2s (Fig. 1c). These cytokines are often associated with protective effects on the development of insulin resistance, as IL-13 production promotes an AAM phenotype

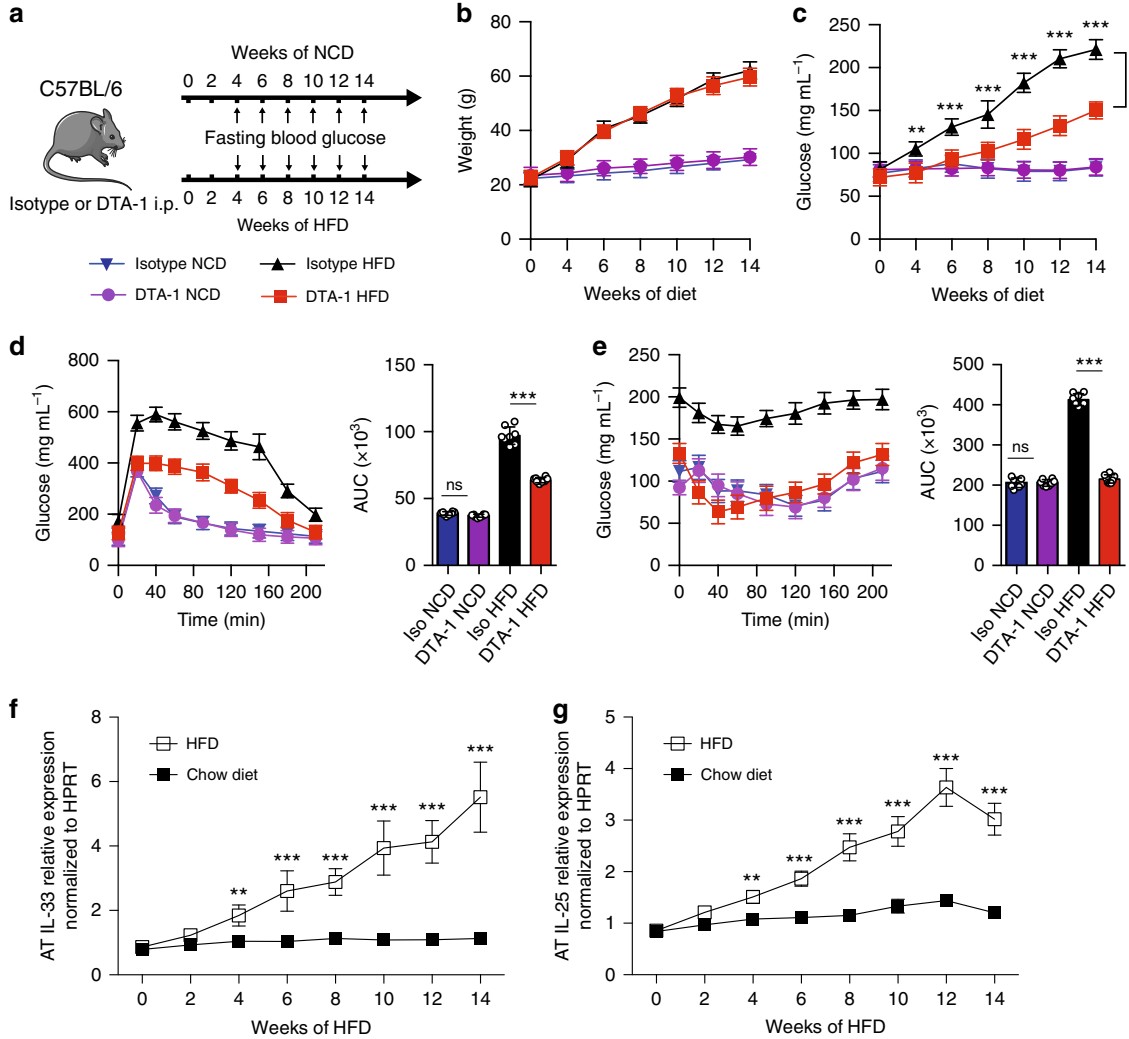

**Fig. 2** Engagement of GITR protects from the onset of Type 2 Diabetes. **a** A cohort of C57/BL6 mice fed a normal chow diet (NCD) or a high fat diet (HFD) were either treated with DTA-1 (1 mg/mouse) or isotype control by intraperitoneal injections every four days according to the scheme, $n = 8$. **b** Total weight and **c** fasting blood glucose levels were measured every two weeks for 14 weeks. Glucose tolerance test (**d**) and insulin tolerance test (**e**) were performed in a cohort of C57/BL6 mice fed a NCD or a HFD and treated for 14 weeks with or without DTA-1. The corresponding area under the curve was calculated for each group. Time-kinetic qPCR expression of IL-33 (**f**) and IL-25 (**g**) on VAT lysates. Error bars are the mean ± SEM. Student's t-test, **$p < 0.01$, ***$p < 0.001$, ns, non-significant. Mouse image provided with permission from Servier Medical Art

and IL-5 is required for eosinophil recruitment and activation[26]. In activated VAT, eosinophils furthermore contribute to AAM maintenance and systemic insulin sensitivity[10]. We therefore measured in vivo intracellular cytokine secretion of VAT-derived ILC2 and M2 macrophage polarization in *Rag2-/-* mice fed HFD for 14 weeks treated with DTA-1 or the isotype control. We first observed that DTA-1 treatment increased the overall presence of ILC2s in the VAT (Fig. 4b), consistent with our findings using C57BL/6 mice (Supplementary Figure 1b-c) and the Ob/Ob mouse model (Supplementary Figure 2f-g). Furthermore, the frequencies of ILC2s that secrete IL-5 (Fig. 4c, top panel) or IL-13 (Fig. 4c, bottom panel) in the VAT are also increased in DTA-1 treated mice as compared to the isotype control group (Fig. 4 and Supplementary Figure 5a). Interestingly, we observed no effect of GITR engagement on IL-5 and IL-13 secretion in ILC2s isolated from the VAT of mice fed a chow diet (Supplementary Figure 5b). Strikingly, the effects were associated with an increased frequency and number of VAT-associated AAMs in DTA-1 group in comparison to the isotype control treated group (Fig. 4f). We next confirmed our previous histology results (Fig. 3j) by flow cytometry. In response to GITR engagement, we

found significantly less CD45[+] cells in the stromal vascular fraction (SVF) of the VAT (Fig. 4a). Furthermore, the total number of macrophages per gram of VAT (Fig. 4e) was also decreased; macrophages were gated as CD45[+] CD11b[hi] F4/80[hi] (Fig. 4d). Taken together, these data suggest that GITR engagement drives Th2 cytokine secretion in ILC2s, which in turn favors an AAM phenotype in the VAT.

**GITR agonist ameliorates established insulin resistance**. As our findings showed that DTA-1 treatment improved the regulation of glucose homeostasis and decreased adiposity in the context of obesity, we next investigated whether DTA-1 treatment could also have a therapeutic effect on mice with established insulin resistance. *Rag2-/-* mice were fed a HFD for 8 weeks to establish insulin resistance after which they were either treated with DTA-1 (1 mg/mouse) or the isotype control by intraperitoneal injections every four days, as described in Fig. 5a for 6 weeks while continuing on the HFD. Similar to our previous results, DTA-1 was associated with increased glucose tolerance and insulin sensitivity compared to isotype control treated mice (Fig. 5b, c, e, f).

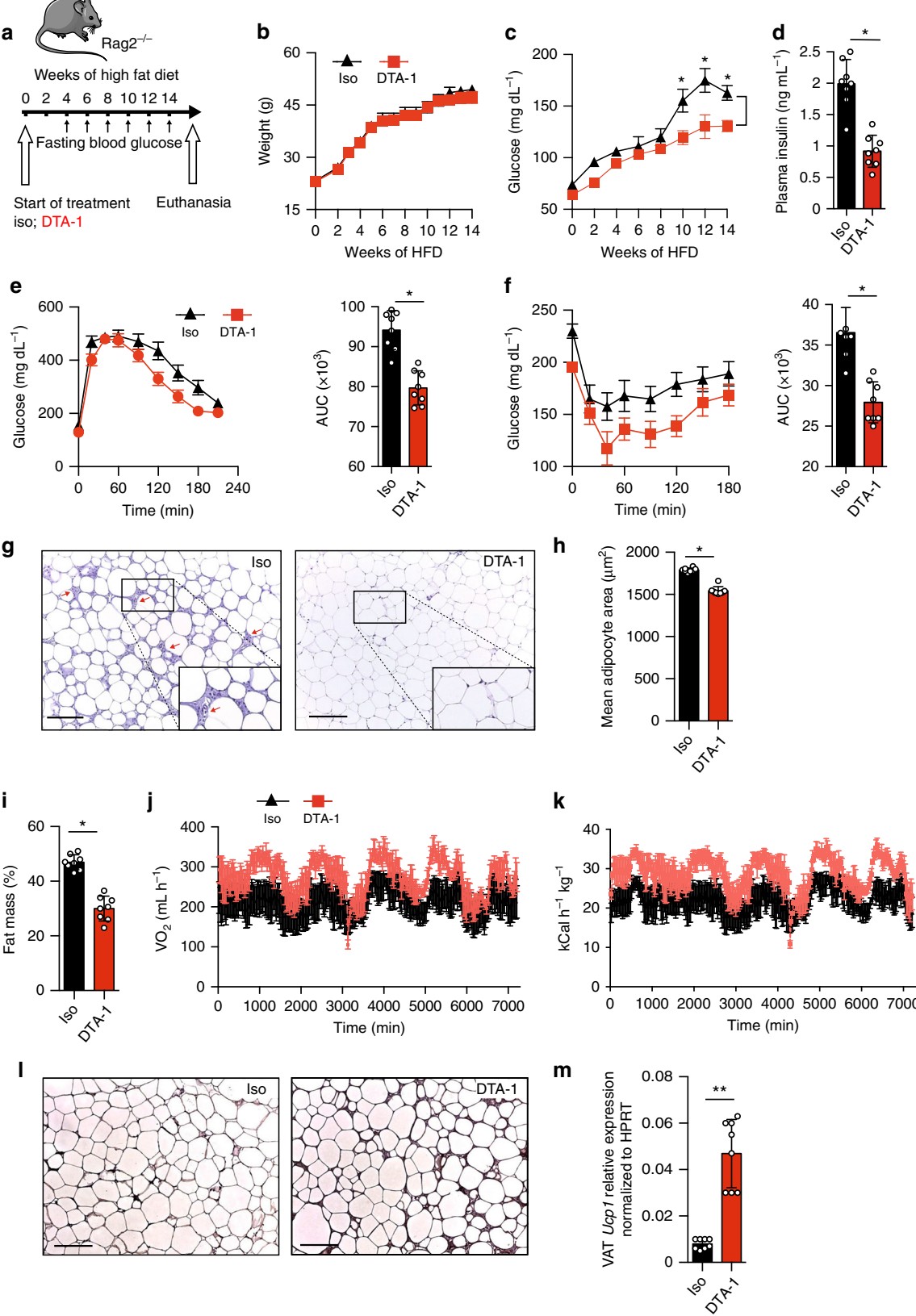

**Fig. 3** Preventive GITR engagement protects from Type 2 Diabetes in *Rag2*[-/-] mice. **a** A cohort of *Rag2*[-/-] mice were fed on HFD and either treated with DTA-1 (1 mg/mouse) or isotype control by intraperitoneal injection every four days according to the scheme, *n* = 8. **b** Total weight and **c** blood glucose levels were measured every two weeks for 14 weeks. **d** Plasma insulin concentration was measured by ELISA. **e** Glucose tolerance test and **f** insulin tolerance test were performed in a cohort of mice after 14 weeks of treatment. The corresponding area under the curve was calculated for each group. **g** Hematoxylin and eosin-stained epididymal adipose tissue sections (x400). Red arrows point to recruited inflammatory cells around adipocytes, scale bars, 100 μm. **h** Corresponding quantitation presented as the mean adipocyte area. **i** Percentage of fat mass was measured using a body composition analyzer. **j, k** CLAMS analysis was performed using individually housed groups of DTA-1 or isotype control treated *Rag2*[-/-] mice maintained on a HFD. **j** Variations in the absolute oxygen consumption and **k** energy expenditure normalized by total body weight were measured. **l** Uncoupling protein 1 (UCP1) immunohistochemistry (IHC) in epididymal adipose tissue sections (x400), scale bars, 100 μm after 14 weeks of treatment and **m** *Ucp1* transcripts levels in VAT lysates, *n* = 8. Error bars are the mean ± SEM. Student's *t*-test, *\*p* < 0.05, *\*\*p* < 0.01. Mouse image provided with permission from Servier Medical Art

We also observed decreased plasma insulin levels in response to DTA-1 treatment (Fig. 5d). Histological analysis of VAT from DTA-1 treated mice also demonstrated less infiltration of leukocytes and a smaller average adipocyte size compared to isotype control-treated mice (Fig. 5g, h). We also observed that DTA-1 treated mice had decreased VAT weights compared to isotype treated mice (Fig. 5i). These data indicate that GITR agonist treatment is able to reverse insulin resistance in mice with established metabolic syndrome.

**GITR engagement is sufficient to prevent insulin resistance.** To assess further if engagement of GITR on ILC2s is sufficient to regulate glucose and prevent induction of insulin resistance, *GITR*[-/-] mice were adoptively transferred with ILC2s isolated from Wild-Type (WT) mice. After adoptive transfer, mice were treated with isotype control or DTA-1 (1 mg/mouse) by intraperitoneal injections every four days, fed a HFD for 14 weeks, and development of insulin resistance was measured in each group (Fig. 6a). DTA-1 treatment did not have any effect on weight gain or improved the glucose tolerance in *GITR*[-/-] mice, which did not receive any WT ILC2s (Fig. 6b–d). However, DTA-1 treatment reduced fasting glycemia and improved glucose tolerance only in *GITR*[-/-] mice that were adoptively transferred with WT ILC2s (Fig. 6b–d). These results demonstrate that the protective effect of DTA-1 treatment is dependent on the expression of GITR on ILC2s. To confirm that the deletion of GITR had no effect on glucose and ILC2 homeostasis in lean mice, WT and *GITR*[-/-] mice were fed NCD for 10 weeks. We performed a glucose tolerance test and assessed VAT ILC2 response in these mice. As expected, we observed no difference on total weight, glucose tolerance, VAT weight, VAT ILC2 numbers and their secretory functions (Supplementary Figure 6a–e). Based on our observations that DTA-1 treatment upregulated intracellular IL-5 and IL-13 expression within VAT ILC2s (Supplementary Figure 1d and Fig. 4), we next investigated whether the protective effect of DTA-1 was mediated by these cytokines. *GITR*[-/-] mice were injected with ILC2s isolated either from WT, IL-5[-/-] or IL-13[-/-] mice. After adoptive transfer, mice were treated with isotype control or DTA-1 by intraperitoneal injections every four days, fed a HFD for 14 weeks, and development of insulin resistance was measured in each group (Fig. 6e). Although we observed no difference on weight gain between all groups of mice, the protective effect of ILC2s on glucose tolerance at the end of treatment disappeared in mice adoptively transferred with IL-13[-/-] ILC2s as compared to WT ILC2s (Fig. 6f, h). Furthermore, the protective effect of GITR engagement on fasting glucose during treatment was either partially or completely repressed in mice injected with either IL-5[-/-] or IL-13[-/-] ILC2s, respectively (Fig. 6g). Collectively, these results suggest that the protective effects of GITR engagement is dependent on ILC2-derived Th2 cytokine secretion and IL-13 in particular.

**GITR engagement induces NF-κB pathway signaling in ILC2s.** To investigate the molecular mechanisms associated with the protective effects of GITR engagement, we next analyzed the gene expression profile of ILC2s either treated with isotype control or DTA-1 (5 μg/mL) for 24 h in vitro, by performing RNA-sequencing (RNA-seq) analysis. RNA sequencing allows to assess the presence and quantity of RNA transcripts in samples. The effect of DTA-1 treatment on whole ILC2 transcriptome is shown as a volcanic plot based on the *p*-value and expression fold-change (FC) of each analyzed gene (Fig. 7a). In red are the statistically significant genes either upregulated or downregulated by a 1.5 FC in response to DTA-1 treatment, also further depicted in a heat plot (Fig. 7b). The cytokine and cytokine receptor genes were correlated based on their modulation in response to DTA-1. As demonstrated in Fig. 7c, expression of IL-5, IL-9, IL-13, and Csf2 were all induced after DTA-1 treatment. These results are consistent with our in vitro (Fig. 1c) and in vivo (Fig. 4c) data. The NF-κB pathway was described as being downstream of the activation pathways of GITR[27]. Consistent with these reports, further analysis of our dataset with Ingenuity Pathway Analysis (IPA) revealed that critical genes involved in the NF-κB pathway were induced in response to DTA-1 treatment compared to the isotype control group (Fig. 7d). This NF-κB pathway was in fact significantly enriched (*p*-value = $5.92 \times 10^{-4}$ and *z*-score = 1.000). Even though the expression of the activated NF-κB p65 subunit (also known as RelA), a crucial gene in NF-κB pathway activation was not induced at the mRNA level, we observed by cytometry at the protein level an upregulation of its expression in response to DTA-1 treatment (Fig. 7e, f). This discrepancy could be explained by the fact that RelA is mainly regulated at the post-translational level by phosphorylation. Further pathway analysis also revealed a network of genes that altogether statistically ($p = 4.13 \times 10^{-9}$) inhibited (*z*-score = −0.929) the apoptosis of leukocytes in response to DTA-1 treatment (Fig. 7g). To assess further the effect of GITR engagement on apoptosis, we quantified by cytometry the percentage of early (AnV[+] 7AAD[−]) and late (AnV[+] 7AAD[+]) apoptotic VAT ILC2s in *GITR*[-/-] versus C57BL/6 control mice. We observed that in absence of GITR expression, ILC2s from the VAT were more apoptotic compared to control mice (Fig. 7h). Collectively, these results demonstrate that GITR engagement consistently induces the secretion of ILC2 effector cytokines, as well as activating the NF-κB downstream pathway and inhibiting ILC2 apoptosis, altogether favoring ILC2 survival and activation.

**GITR on human ILC2s and Th2 cytokine production.** We next explored whether human ILC2s express GITR and whether GITR engagement could play a crucial role in the activation and function of human ILC2s. Purified peripheral blood ILC2s from healthy donors were cultured with recombinant human (rh) IL-2, rhIL-7, and rhIL-33 and plate bound GITR-L-Fc or isotype control for 0, 24, and 48 h (Supplementary Figure 7a). We found that GITR was induced on human ILC2s in a time-dependent

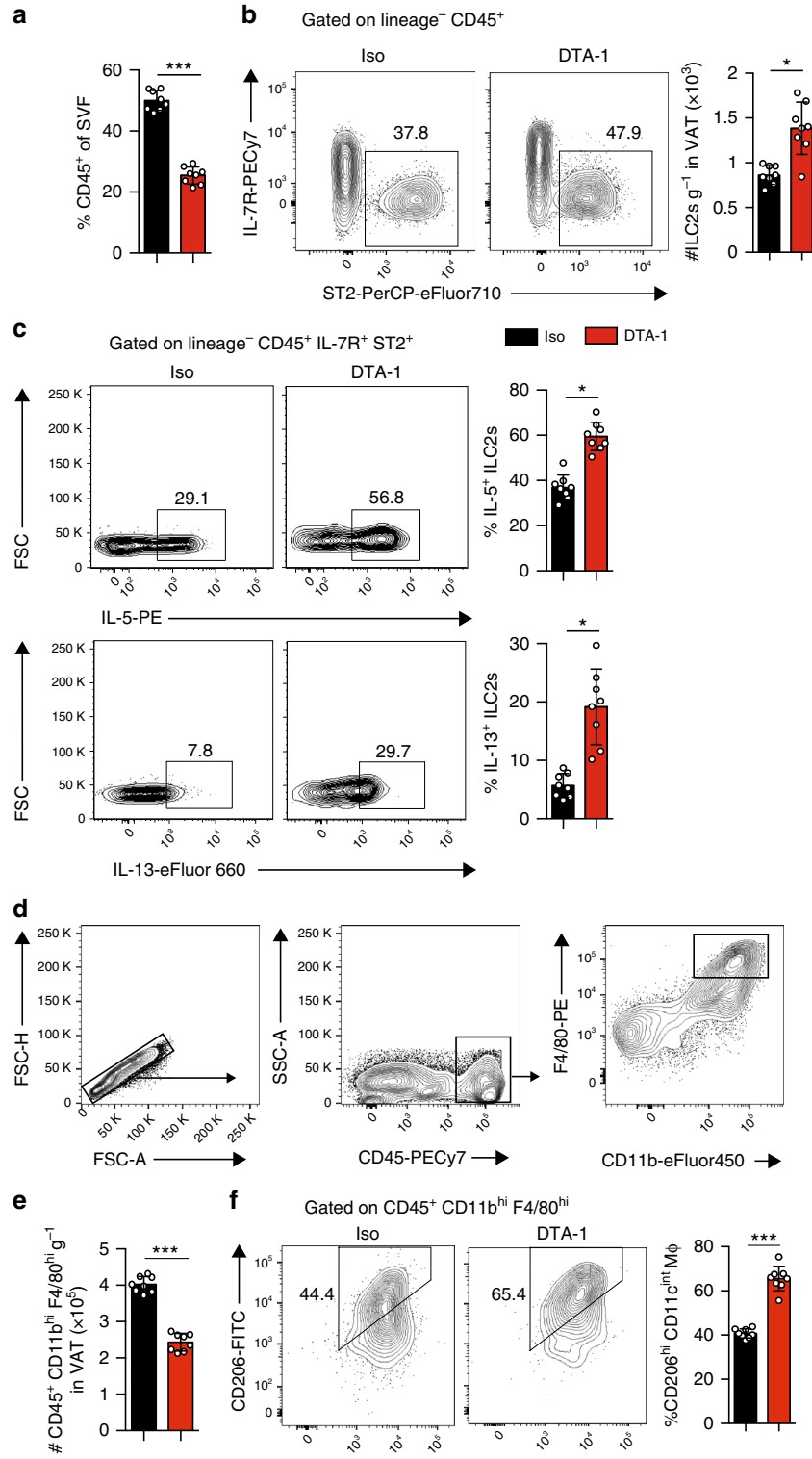

**Fig. 4** GITR engagement drives ILC2-derived secretion and M2 macrophage polarization. A cohort of *Rag2*$^{-/-}$ mice were fed on HFD and either treated with DTA-1 (1 mg/mouse) or isotype control by intraperitoneal injection every four days, $n = 8$. **a** The frequency of CD45$^+$ cells of the VAT stromal vascular fraction (SVF) was quantified by flow cytometry. **b** Representative flow cytometry plots of VAT Lin$^-$CD45$^+$IL-7R$^+$ST2$^+$ ILC2s in each group after 14 weeks of treatment, and corresponding quantitation presented as the number of ILC2s per gram of VAT. **c** Representative flow cytometry plots of intracellular IL-5 (top panel) and IL-13 (bottom panel) in VAT ILC2s and corresponding quantitation after 14 weeks of treatment, presented as frequency of positive ILC2s. **d** Gating strategy of macrophages in mouse VAT as CD45$^+$ CD11b$^{hi}$ F4/80$^{hi}$. **e** Quantification of the number of CD45$^+$ CD11b$^{hi}$ F4/80$^{hi}$ macrophages per gram of VAT. **f** Representative flow cytometry plots of CD45$^+$CD11b$^{hi}$F4/80$^{hi}$CD206$^+$CD11c$^+$ M2 macrophages in VAT and corresponding quantitation after 14 weeks of treatment. Error bars are the mean ± SEM. Student's *t*-test, *$p < 0.05$, ***$p < 0.001$

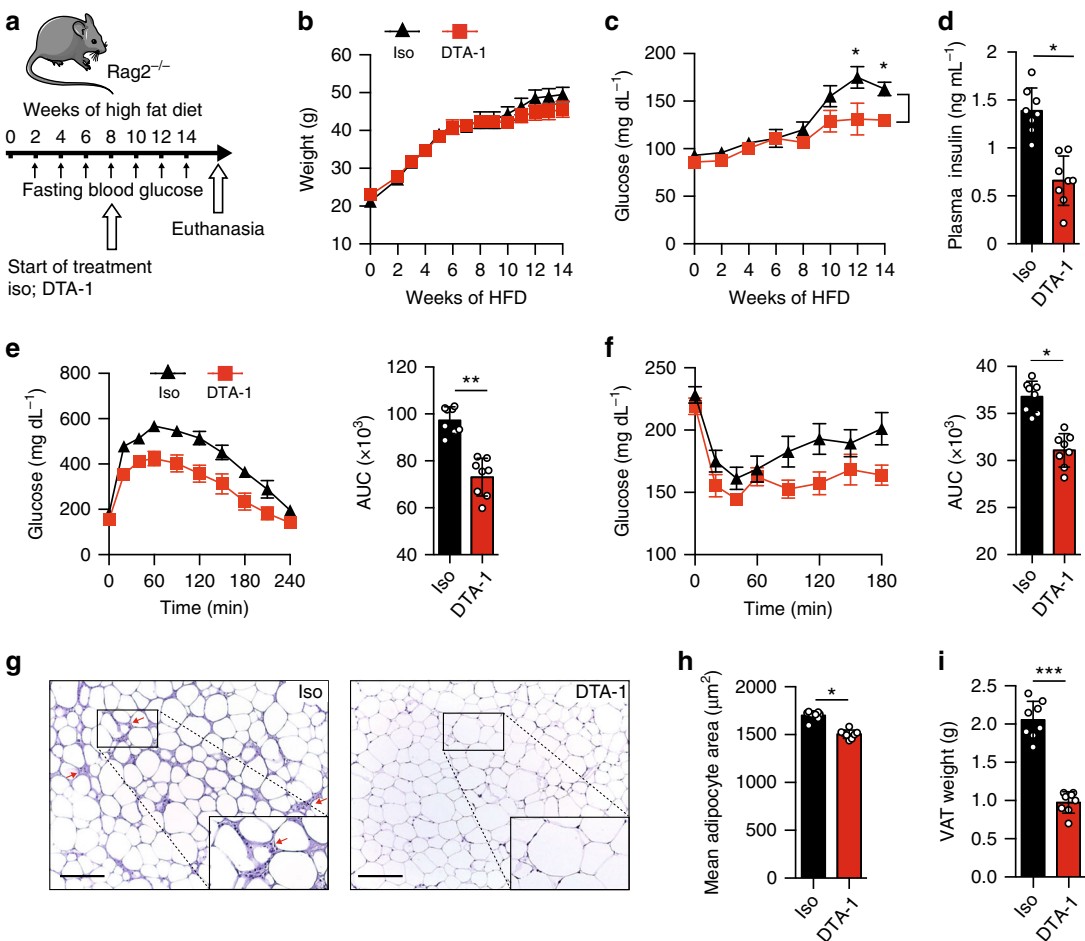

**Fig. 5** Therapeutic GITR treatment ameliorates established Type 2 Diabetes in *Rag2*[-/-] mice. **a** A cohort of *Rag2*[-/-] mice were fed on HFD for 14 weeks. After 8 weeks, mice were either treated with DTA-1 (1 mg/mouse) or isotype control by intraperitoneal injection every four days according to the scheme, $n = 8$. **b** Total weight and **c** fasting blood glucose levels were measured every two weeks for 14 weeks. **d** Plasma insulin concentration was measured by ELISA. **e** Glucose tolerance test and **f** insulin tolerance test were performed in a cohort of mice after 8 weeks of treatment. The corresponding area under the curve was calculated for each group. **g** Hematoxylin and eosin-stained epididymal adipose tissue sections (×400). Red arrows point to recruited inflammatory cells around adipocytes, scale bars, 100 μm. **h** Corresponding quantitation presented as the mean adipocyte area. **i** Weight of the VAT (epididymal adipose tissue) after 8 weeks of treatment. Error bars are the mean ± SEM. Student's t-test, $p < 0.05$, **$p < 0.01$, ***$p < 0.001$. Mouse image provided with permission from Servier Medical Art

manner. Indeed, the longer the incubation in presence of GITR-L-Fc, the stronger was GITR expression on the cell surface (Fig. 8a). In line with these observations, human GITR was also expressed on the surface of VAT-derived ILC2s from healthy subjects (Fig. 8b and Supplementary Figure 7b). Considering that upon activation ILC2s secrete higher levels of Th2 cytokines such as IL-5 and IL-13, we also measured by Luminex Th2 cytokines in the supernatant of human blood ILC2s cultured on plate bound GITR-L-Fc for 24 h. In line with our data obtained with murine ILC2s, in response to GITR engagement human ILC2s also secrete high amounts of IL-5, IL-13, GM-CSF, IL-8, and IL-9 in presence of rhIL-33 (Fig. 8c). Taken together, these results show that human ILC2s express GITR and that GITR engagement induces Th2 cytokine secretion on rhIL-33 activated ILC2s, suggesting a co-stimulatory role of the GITR receptor in human ILC2s, similar to that observed in mice.

## Discussion

In this report, we demonstrate that GITR—a member of the TNFR superfamily and known as a co-stimulatory molecule—is expressed on both murine and human ILC2s from the VAT. We

discovered that GITR engagement on ILC2s with the specific agonist DTA-1 induces Th2 cytokine secretion in activated ILC2s but has no effect on naïve ILC2s. Importantly, we showed that GITR engagement is protective against insulin resistance onset and can also ameliorate established insulin resistance. By inducing Th2 cytokine secretion in activated ILC2s, GITR engagement modulates macrophage polarization, which in turn favors insulin sensitivity. Our results suggest the potential of GITR engagement not only as a therapeutic molecule against insulin resistance but also as an immune checkpoint for activated ILC2s.

GITR as a Glucocorticoid-induced tumor necrosis factor receptor-related protein is highly expressed in activated T cells and regulatory T cells (Tregs)[28]. This receptor was initially cloned from a mouse T-cell hybridoma as a dexamethasone-inducible molecule[29]. GITR plays a crucial role in the differentiation of thymic Tregs (tTregs) and in the proliferation of both tTregs and peripheral Tregs. GITR has been described as a key marker of functional Tregs and participates in costimulation of effector T cells (Teffs). The costimulatory effect of GITR engagement in T cells induces T cell expansion and cytokine secretion[22,30,31]. This T cell activation was associated with enhanced cell cycle progression, cytokine production, such as IL-2 and intracellular

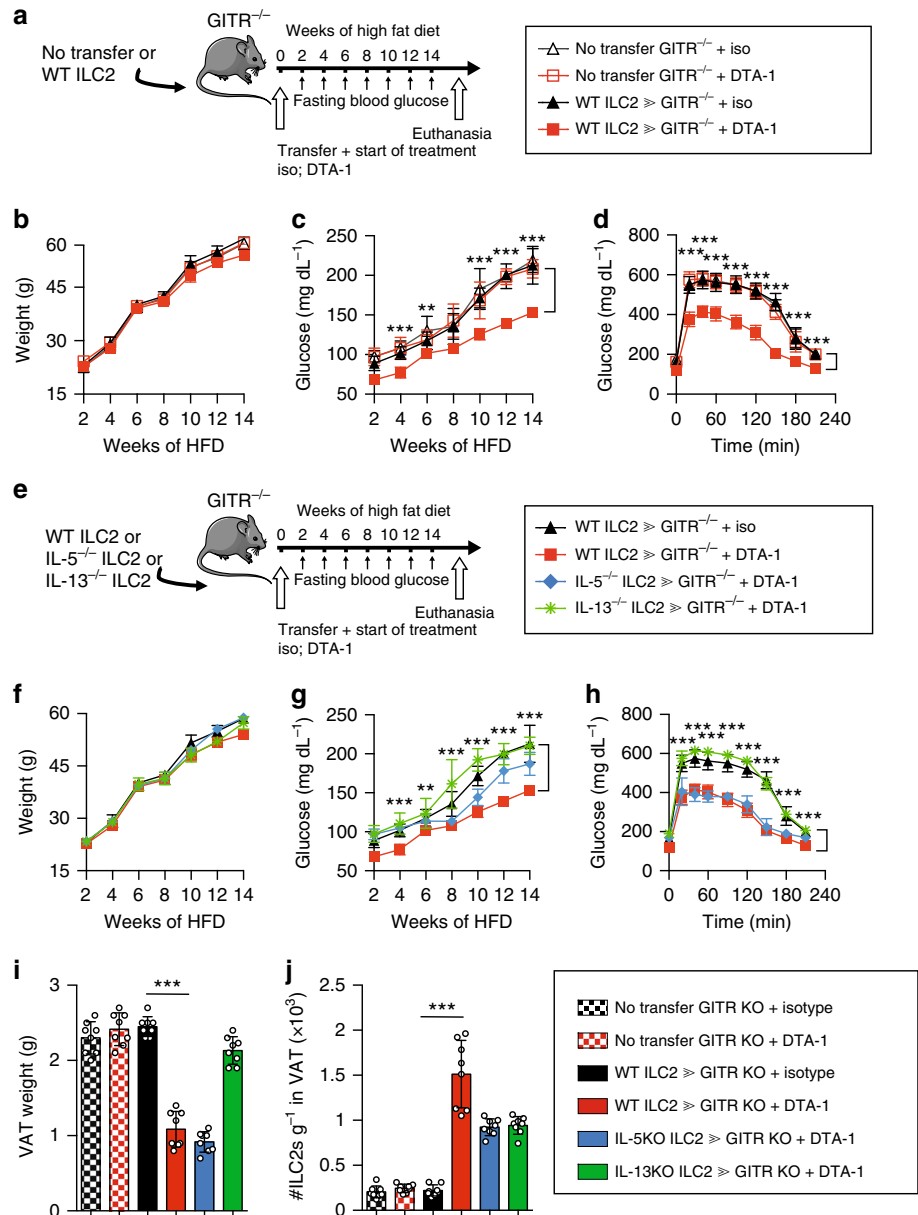

**Fig. 6** GITR engagement effects are dependent on GITR on ILC2s and IL-13. **a** A cohort of *GITR⁻/⁻* mice were adoptively transferred or not with ILC2s from WT mice. After the adoptive transfer, mice were treated with DTA-1 (1 mg/mouse) or isotype control and fed on HFD for 14 weeks according to the scheme, $n = 8$. **b** Total weight and **c** fasting blood glucose levels were measured every two weeks for 14 weeks. **d** Glucose tolerance test was performed after 14 weeks of treatment. **e** A cohort of *GITR⁻/⁻* mice were adoptively transferred with ILC2s from either WT, IL-5⁻/⁻, or IL-13⁻/⁻ mice. After the adoptive transfer, mice were treated with DTA-1 or isotype control and fed on HFD for 14 weeks according to the scheme, $n = 8$. **f** Total weight and **g** fasting blood glucose levels were measured every two weeks for 14 weeks. **h** Glucose tolerance test was performed after 14 weeks of treatment. VAT weight (**i**) was measured and number of ILC2s per gram of VAT (**j**) was quantified 14 weeks after adoptive transfer. Error bars are the mean ± SEM. Student's *t*-test, **$p < 0.01$, ***$p < 0.001$. Mouse image provided with permission from Servier Medical Art

signaling including activation of NF-κB pathway[30]. Similarly, other groups showed that Treg cell proliferation triggered by GITR costimulus is linked to the loss of the anergic phenotype and suppressor activity[22]. In addition, GITR signaling can also induce proliferation of antigen-stimulated T cells when exposed to their specific antigen[31]. These previous findings suggest that GITR can act as a costimulatory molecule on Teff cells.

In our study we observed that activation of GITR signaling pathway resulted in enhanced cytokine secretion, increased ILC2 recruitment and induction of a network of genes inhibiting leukocyte apoptosis suggesting that GITR engagement also promotes cell survival. These results clearly illustrate that in the context of

metabolic syndrome, ILC2s are primed and readily respond to GITR costimulation. This primary signal is critical for GITR signaling as GITR engagement has no effect of naïve ILC2s. The increased concentration of IL-33 and IL-25 in the VAT observed in our study in association with the low-grade inflammation linked to metabolic syndrome constitute a primary signal vital for ILC2 activation. Furthermore, the lack of intracellular IL-5⁺ or IL-13⁺ induction in VAT ILC2s in response to GITR engagement in lean mice further highlights the requirement for a primary signal in GITR signaling. Our results clearly indicate that in addition to T cells, GITR can act as an immune checkpoint capable of costimulating activated ILC2s. In this regard, various

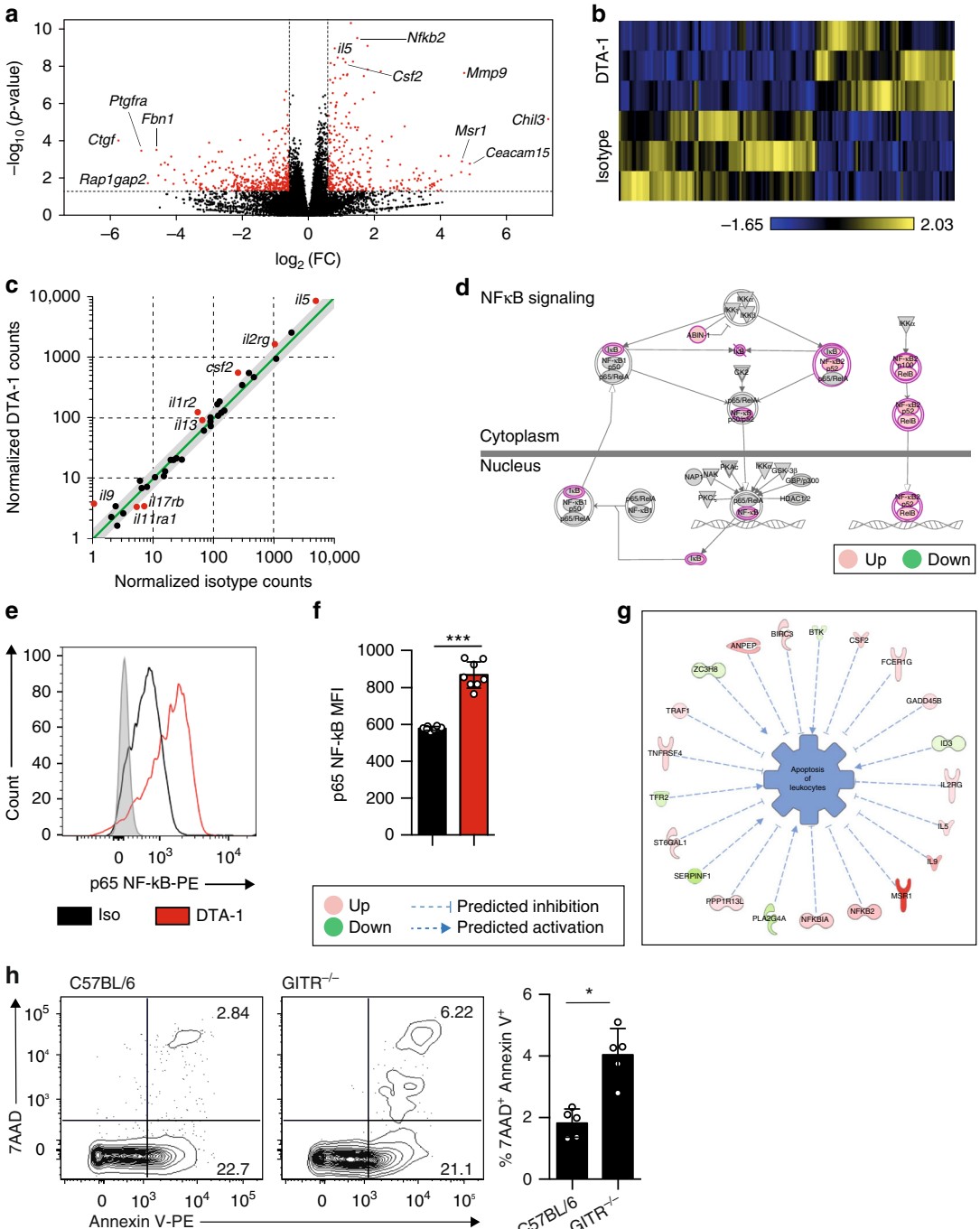

**Fig. 7** GITR engagement induces NF-κB pathway signaling in ILC2s. **a** Volcano plot comparison of whole transcriptome gene expression of sorted WT ILC2s from VAT treated with isotype control or DTA-1 (5 μg/mL) for 24 h in vitro, $n = 3$. Differentially expressed genes (defined as statistically significant adjusted $p$-value < 0.05) with changes of at least 1.5 fold-change (FC) are shown in red. Notable differentially expressed genes are labeled. **b** Heat plot of differentially expressed genes. **c** Selected genes plotted as the normalized counts in isotype control-treated compared to DTA-1-treated mice. Differentially expressed genes (defined as in a) are shown in red. Gray area indicates region of 1.5 fold-change cutoff or lower change in expression. Shown are expression of cytokine and cytokine receptor genes. **d** Upregulated (red) and downregulated (green) genes in the NF-κB pathway. **e** Representative histogram of the expression of NF-κB p65 in isolated ILC2s from mice challenged with IL-33 and cultured in vitro for 24 h with DTA-1 (red) or isotype control (black). The level of isotype-matched stain control is shown as a gray-filled histogram. **f** Corresponding quantification presented as Mean Fluorescence Intensity of NF-κB p65 with DTA-1 or isotype control. **g** Network analysis of upregulated (red) and downregulated (green) genes overall significantly predicted to inhibit the apoptosis of leukocytes. **h** Representative flow cytometry plots of VAT Lin⁻CD45+IL-7R+ST2+Annexin V⁺ 7-AAD⁺ ILC2s from C57/BL6 or *GITR*⁻/⁻ mice ($n = 5$) and corresponding quantitation presented as the percentage of ILC2s. Error bars are the mean ± SEM. Student's *t*-test, *$p < 0.05$, ***$p < 0.001$

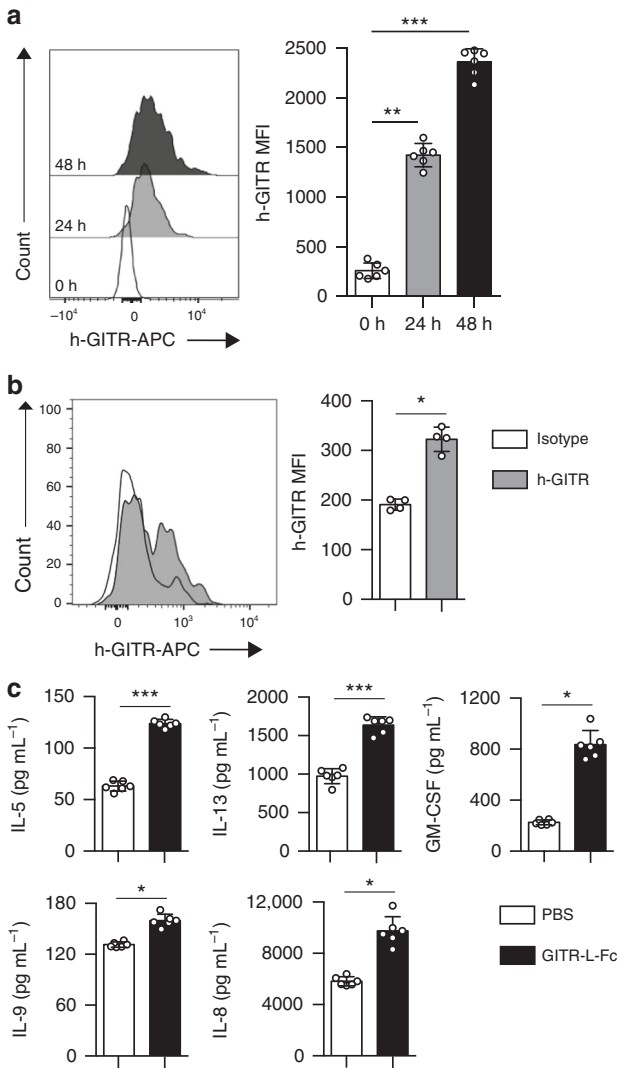

**Fig. 8** GITR is expressed on human ILC2s and augments Th2 cytokine production. FACS-sorted human blood ILC2s were cultured with recombinant human (rh) IL-2, rhIL-7, and rhIL-33 and plate bound GITR-L-Fc or isotype control, human IgG, n = 6. **a** Representative FACS plots of h-GITR expression after 0, 24, and 48 h of stimulation on ILC2s and corresponding quantitation presented as MFI. **b** Representative FACS plot of h-GITR staining on VAT-ILC2s isolated from healthy subjects and corresponding quantitation, n = 4. **c** The levels of IL-5, IL-13, GM-CSF, IL-8, and IL-9 in the culture supernatants were measured by Luminex after 24 h of stimulation. Data representative of 6 individual blood donors, n = 6. Error bars are the mean ± SEM. Student's t-test, *p < 0.05, **p < 0.01, ***p < 0.001

types of immune cells are recruited to VAT as a result of the low-grade inflammation associated with obesity, among them are monocytes/macrophages. For example, through the production of different types of inflammatory mediators, macrophages acquire a phenotype that favors the resolution of the inflammation. Alternatively activated macrophages are a subset of anti-inflammatory macrophages also identified as "M2" macrophages, and are characterized by IL-10 secretion, arginase 1 and CD206 (mannose receptor) expressions[32]. They are vital components of immune responses as the decrease of AAM numbers in the adipose tissue has been demonstrated to promote insulin resistance, highlighting their protective role against metabolic disease. Furthermore, AAMs have also been described as key players in adipose

tissue remodeling and in regulating classical inflammation[33]. In the adipose tissue, the activation of ILC2s, notably by IL-33, is associated with increased numbers of AAMs in adipose tissue. In line with this, studies have shown that mice lacking either IL-33 or ST2 are characterized by reduced ILC2 activation and decreased numbers of AAMs in adipose tissue[4,6,34]. Similarly, IL-33 treatment is metabolically beneficial as it has been associated with decreased adiposity, induced beiging of fat, improved fasting glycemia and insulin tolerance in obese mouse models[6,35,36]. In our study, in response to GITR engagement in ILC2s the number of AAMs in the adipose tissue was increased. This polarization of AAMs was associated with decreased insulin resistance, increased glucose tolerance and reduced adiposity. These findings are consistent with metabolic underpinnings of anti-inflammatory mechanisms in other obesity studies.

ILC2s coordinate innate type 2 immune responses through secretion of key Th2-associated cytokines like IL-5 and IL-13. Interestingly, it has been described that IL-5 plays an important role in the accumulation of eosinophils which in turn play a critical role in the maintenance of AAMs[26]. However, the migration of eosinophils into the adipose tissue and the reconstitution of AAMs were shown to be dependent on IL-13[10]. Similarly, in absence of IL-5 or IL-13, metabolic syndrome is exacerbated in mice fed a HFD. Inversely, mice overexpressing IL-5 or treated with IL-13 are leaner and more glucose tolerant[37]. Our results are consistent with these observations since we show that GITR engagement induces Th2 cytokine secretion in activated ILC2s, resulting in increased numbers of AAMs in the adipose tissue and improvement of glucose tolerance in mice. We further validated that the protective role of GITR engagement was dependent on Th2 cytokine secretion by using transgenic knockout mouse models in conjunction with adoptive transfer studies. Interestingly our results demonstrated that the protective effect of GITR engagement on insulin resistance was critically dependent on IL-13 secretion. The secretion of IL-13 has been previously described to polarize macrophages towards an "alternatively activated" phenotype, which in turn helps to maintain glucose homeostasis and dampen inflammation[10]. It is noteworthy to mention that GITR engagement demonstrated not only a preventive role against the onset of insulin resistance but also therapeutic effects, as the binding of the GITR specific agonist on ILC2s improved insulin sensitivity of diabetic mice. This notion is consistent with our in vitro results and recent epidemiological studies. For example, of 26 inflammatory biomarkers evaluated in a longitudinal analysis of ~1000 subjects from the Rotterdam Study, plasma IL-13 levels were inversely associated with progression from normoglycemia to pre-diabetes, incident T2DM, and initiation of insulin therapy[38].

In a study by Esparza et al., GITR engagement was found to induce the NF-κB pathway in lymphocytes[27] and in another study, it was suggested that upon activation with IL-33, the downstream canonical NF-κB signaling pathway is activated in target cells resulting in Th2 cytokines production[39]. Our results also address this molecular mechanism for GITR engagement on ILC2s since we demonstrated that the NF-κB pathway was significantly enriched in our transcriptomic analysis, and the expression of the activated NF-κB p65 subunit at the protein level was induced in response to DTA-1 treatment. Collectively, these results suggest that GITR engagement could repress a key pro-inflammatory signaling pathway mediated via NF-κB.

We further investigated whether this protective effect of GITR engagement was relevant in human cells. We observed that upon activation ILC2s express GITR in a time-dependent manner resulting in an induction of Th2 cytokine secretion. In accordance with the effects seen in murine ILC2s, specific GITR agonist significantly stimulated ILC2s suggesting a costimulatory role of

the GITR receptor in human ILC2s. Taken in their entirety, our results suggest that activating GITR signaling pathway might be an effective therapeutic strategy to prevent and improve T2DM.

In conclusion, in this study we demonstrate GITR expression on ILC2s in the VAT, and that in activated ILC2s upon engagement this pathway results in Th2 cytokine secretion, altogether demonstrating a co-stimulatory immune checkpoint role for GITR. Strikingly, we also demonstrated that GITR engagement improves glucose tolerance and insulin sensitivity not only in preventive but also in a therapeutic manner. This protective role of GITR engagement is dependent on an enhanced Th2 cytokine production. Therefore, our results suggest a protective role of GITR signaling in the metabolic syndrome and present the specific GITR agonist as a preventive and therapeutic candidate for regulating T2DM.

## Methods

**Mice**. GITR deficient mice were obtained from Dr. Tania Watts (University of Toronto, Toronto, Canada) and Dr. Carlo Riccardi (University of Perugia, Perugia, Italy). C57BL/6J, Ob/Ob (B6.Cg-Lep[ob]/J), RAG2 deficient (C.B6(Cg)-Rag2[tm1.1Cgn]/J), IL-13 deficient (C.129P2-Il13[tm1.1Anjm]) and IL-5 deficient (C57BL/6-Il5[tm1Kopf]/J) mice were purchased from the Jackson Laboratory (Bar Harbor, Maine). All mice were bred in our animal facility at the Keck School of Medicine, University of Southern California (USC). Four to eight-week-old aged and sexed-matched mice were used in the studies. All animal studies were approved by the USC institutional Animal Care and Use Committee and conducted in accordance with the USC Department of Animal Resources' guidelines.

**Diet-induced obesity and in vivo treatments**. When indicated, mice were fed a high fat diet (HFD, Rodent diet with 60 kcal% Fat, D12492i) from Research Diets Inc. (New Brunswick, New Jersey) for the indicated times, as described before[40]. All other mice were fed a normal chow diet (NCD). For in vivo experiments investigating the effect of GITR engagement, GITR agonist DTA-1 (1 mg/mouse, BioX-Cell, West Lebanon, New Hampshire, BE0063), or the monoclonal antibody rat IgG2b isotype control (BioXCell) (1 mg/mouse) was administered intraperitoneally every 4 days from the indicated start of treatment until termination of the experiment.

**In vivo metabolic phenotyping**. To measure weight and fasting blood glucose levels, mice were fasted overnight (~14–16 h), weighed and glucose values were measured using a glucometer (Contour®Next EZ, Bayer, Leverkusen, Germany) collecting a drop of blood every two weeks. For intraperitoneal glucose tolerance tests (ip-GTT), mice were fasted overnight (~16 h), weighed and injected with 2 g/kg 20% D-glucose (Sigma Aldrich) solution intraperitoneally. Blood glucose values were measured for each mouse by collecting a drop of blood before injection and at 20, 40, 60, 90, 120, 150, 180, 210, and 240 min post-injection. For insulin tolerance tests (ITT), mice were fasted for 5 h, weighed and injected with 0.5U/kg human insulin (Novolin®, Novo Nordisk®, Bagsværd, Denmark) diluted in Sodium Chloride Solution 0.9% w/v (Azer Scientific, Morgantown, Pennsylvania) solution intraperitoneally. Blood glucose values were measured for each mouse by collecting a drop of blood before injection and at 20, 40, 60, 90, 120, 150, 180, 210, and 240 min post injection. For both glucose and insulin tolerance tests, the data were analyzed by quantifying the area under the curve (AUC) for each group of mice. When indicated, blood was collected by cardiac puncture and plasma insulin levels were measured using the ultra-sensitive mouse insulin ELISA Kit (Crystal Chem High Performance Assays). Metabolic analysis of whole animals were performed using PhenoMaster/LabMaster home cages following the manufacturer's instructions (TSE Systems). Briefly, at the indicated time after onset of treatment, mice were singly housed and measures were taken every 27 min for 5 days. Measures included oxygen consumption and carbon dioxide output, as variations in oxygen consumption and energy expenditure (heat) over time were calculated. Energy expenditure was normalized to body mass and lean mass. Mouse body composition was assessed with a minispec LF90 TD-NMR (Time-domain nuclear magnetic resonance spectroscopy) analyzer (Bruker) to quantify fat and lean mass.

**Murine ILC2 isolation and in vitro stimulation**. Murine VAT and human peripheral ILC2s were isolated to >95% purity using the FACS Aria III cell sorter. For in vivo stimulation of murine VAT ILC2s, carrier free rm-IL-33 (Biolegend, San Diego, CA, 1 μg/mouse in 200 μL) was administered intraperitoneally to mice on three consecutive days. On day 4, murine ILC2s were isolated based on the lack of expression of classical lineage markers (CD3ε, CD45R, Gr-1, CD11c, CD11b, Ter119, TCRγδ, and FCεRI) and expression of CD45, ST2, and CD117. Isolated ILC2s were stimulated (5 × 10⁴/mL) with rm-IL-2 (10 ng/mL) and rm-IL-7 (10 ng/mL) for 48 h at 37 °C in presence of GITR agonist DTA-1 (5 μg/mL) from BioX-Cell, West Lebanon, New Hampshire (BE0063) or the monoclonal antibody rat IgG2b isotype control (BioXCell). For adoptive transfer experiments, 2.5 × 10⁵

purified ILC2s were adoptively transferred intravenously in 200 μL PBS into the recipients at the start of the indicated treatment.

**Human ILC2 isolation and in vitro stimulation**. All human studies were approved by USC Institutional review board and conducted in accordance to the principles of the Declaration of Helsinki. Informed consent was obtained from all human participants. For human ILC2s from blood, peripheral blood mononuclear cells (PBMCs) were first isolated from human fresh blood by diluting the blood 1:1 in PBS and adding to SepMate™-50 separation tubes (STEMCELL Technologies Inc, Vancouver, Canada) prefilled with 15 mL Lymphoprep™ each (Axis-Shield, Oslo, Norway) and centrifugation at 1200 × g for 15 min. Human ILC2s were then isolated by cell sorting based on the lack of expression of classical lineage markers (CD3, CD14, CD16, CD19, CD20, CD56, CD235a, CD1a, CD123) and expression of CD45, CRTH2 and CD161. Purified human ILC2s were stimulated (5 × 10⁴/mL) with recombinant human (rh)-IL-2 (20 ng/mL), rh-IL-7 (20 ng/mL) and rh-IL-33 (20 ng/mL) for the indicated times at 37 °C in presence or absence of plate-bound GITR-L-Fc obtained from Dr. Alan Epstein at USC as described previously[41]. For human ILC2s from adipose tissue, adipose tissue samples were digested in collagenase IV (MP Biomedicals, LLC) at 37 °C for one hour and then processed on a 70 μm nylon cell strainer (Falcon®) into a single cell suspension.

**Supernatant cytokine measurement**. Human IL-5 ELISA MAX™ Deluxe was purchased from BioLegend, Ready-SET-Go!®. ELISA for human IL-13, mouse IL-5, and IL-13 were purchased from ThermoFisher Scientific and the level of cytokines were measured according to the manufacturer's instructions. Other cytokines were measured by multiplexed fluorescent bead-based immunoassay detection (MILLIPLEX® MAP system, Millipore Corporation, Missouri U.S.A.) according to the manufacturer's instructions, using a combination of 32-plex (MCYT-MAG70KPX32) and 41-plex (HCYTMAG-60K-PX41) Millipore Human Cytokine panel kits. For each assay, the curve was performed using various concentrations of the cytokine standards assayed in the same manner and analyzed using MasterPlex2012 software (Hitachi Solutions America, Ltd.), as described by our group before[42,43].

**Tissue preparation and flow cytometry**. Epididymal adipose tissue used as representative VAT and BM were collected at the indicated times after transcardial perfusion to clear organs of red blood cells. VAT was processed to single cell suspensions as previously described[44], and BM cells were collected by flushing bones with PBS. Stained cells were analyzed on FACSCanto II and/or FACSARIA III systems (Becton Dickinson) and the data were analysed using FlowJo version 10 software (TreeStar, Ashland, Oregon). The following mouse antibodies were used:

Biotin-CD3e (100304; clone: 145-2C11; 1/200), Biotin- D45R/B220 (103204; clone: RA3–6B2; 1/200), Biotin-Gr-1 (108404; clone: RB6–8C5; 1/200), Biotin-CD11c (117304; clone: N418; 1/200), Biotin-CD11b (101204; clone: M1/70; 1/200), Biotin-Ter119 (116204; clone: TER-119; 1/200), Biotin—FceRIa (134304; clone: MAR-1; 1/200), FITC-Streptavidin (405202; 1/500), PE-Cy7-CD127 (135014; clone: A7R34; 1/300), APCCy7-CD45 (103116; clone: 30-F11; 1/300), APC/Cy7-CD11c (117314; clone: N418; 1/300), PECy7-CD45 (103114; clone: 30-F11; 1/300), PE-F4/80 (123110; clone: BM8; 1/300), FITC-CD206 (141704; clone C068C2; 1/300), PE-IL-5 (504304; clone: TRFK5; 1/200), PE-Rat IgG1k (cat. #400407; clone: RTK2071; 1/200) were purchased from BioLegend. AlexaFluor647-CD357 (GITR) (565151; clone: DTA-1; 1/200) and AlexaFluor647-Rat IgG2b, k (557691; 1/200) were purchased from BD Pharmingen. eFluor 660-IL-13 (50-7133-82; clone: eBio13A; 1/200), eFluor 660-IgG1k (50-4301-82; clone: eBRG1; 1/200), Biotin-TCR-gd (13-5711-85; clone: eBioGL3; 1/200), PerCP-eFluor710-ST2 (46-9335-82; clone: RMST2-2; 1/300) APC-CD127 (17-1271-82; clone: A7R34; 1/300), eFluor450-CD11b (48-0112-82; clone: M1/70; 1/300) were purchased from eBioscience. Intranuclear staining was performed using the Foxp3 Transcription Factor Staining Kit (ThermoFisher Scientific), according to the manufacturer's instructions. PE anti-human/mouse RelA NFκB p65 (IC5078P, 1/200) and PE-IgG2B (IC0041P, 1/200) were purchased from R&D Systems. Intracellular staining was performed using the BD Cytofix/ Cytoperm kit (BD Bioscience, San Jose, CA), according to the manufacturer's instructions. The following human antibodies were used: FITC- Lineage (348801; 1/100), FITC-CD235a (349104; clone: HI264; 1/500), FITC-FCeRIa (334608; clone: AER-37; 1/100), FITC-CD1a (300104; clone: HI149; 1/100) FITC-CD123 (306014; clone: 6H6; 1/100), APCCy7-CD45 (304014; clone: HI30; 1/ 100), PerCP/Cy5.5-CD161 (339908; clone: HP-3G10; 1/100), APC-CD357 (GITR) (371206; clone: 108-17; 1/100), APC-IgG2ak (400221; clone: MOPC-173, 1/100), PE-CD294 (CRTH2) (350106; clone: BM16; 1/100), PE/Cy7-CD127 (IL-7Ra) (351320; clone: A019D5; 1/100) were purchased from BioLegend.

**RNA Sequencing (RNA-seq) and data analysis**. Freshly isolated ILC2s after 3 i.p. injections of 1 μg rm-IL33 were stimulated (5 × 10⁴/mL) with rm-IL-2 (10 ng/mL) and rm-IL-7 (10 ng/mL) for the indicated times at 37 °C with GITR agonist DTA-1 (5 μg/mL) or the isotype control for 48 h. Total RNA was isolated using Micro-RNAeasy (Qiagen, Valencia, California). 10 ng of input RNA was used to produce cDNA for downstream library preparation. Samples were sequenced on a NextSeq 500 (Illumina) system. Raw reads were aligned, normalized and further analyzed

using Partek® Genomics Suite® software, version 7.0 Copyright ©; Partek Inc., St Louis, MO, USA. Pathway analysis was performed using the Qiagen Ingenuity Pathway Analysis (IPA) software.

**Histologic analysis**. Samples were collected and fixed with 4% paraformaldehyde in PBS. After fixation, tissues were embedded in paraffin, before cutting into 4 μm sections and staining with hematoxylin and eosin (H&E) or performing immunohistochemistry (IHC) with rabbit anti-UCP1 antibody (Abcam, ab10983)[6]. For IHC, rehydrated sections were microwaved in 10 mM citric acid buffer (pH 6.0) for antigen retrieval, and endogenous peroxidases were quenched with BLOXALL Blocking solution (Vector Laboratories, Burlingame, California). Sections were blocked with Avidin D, biotin and protein blocking agent in sequential order followed by application of the anti-UCP1 antibody (1:500). A biotinylated anti-rabbit antibody was used as a secondary antibody. Horseradish peroxidase-conjugated ABC reagent was applied, and then DAB reagent was used to develop the signal before counterstaining in hematoxylin and dehydrating the sections in preparation for mounting. Stained sections were acquired using a Leica DME microscope and Leica ICC50HD camera (Leica, Wetzlar, Germany) and analyzed using Leica LAS EZ software. Adipocyte size quantitation was performed using the Adiposoft plugin of ImageJ (NIH, Maryland, USA).

**Real-time PCR**. Total RNA was prepared using the RNeasy lipid tissue kit (QIAGEN, Hilden, Germany) following the manufacturer's instructions.) Quantitative PCR was performed on a LightCycler 2.0 (Roche Diagnostics, Mannheim, Germany). Relative mRNA levels were determined by comparison with a reference gene using the delta-delta CT method.

The sequences of the gene-specific primers are: *Cidea* (forward primer) CGAGT TTCAAACC-ATGACCGAAGTAGCC, *Cidea* (reverse primer) CTTACTACCCG GTGTCCATTTCTGTCCC, *Cox7a* (forward primer) CTCTTCCAGGCCGACAA TGACCTC, *Cox7a* (reverse primer) G-CCCAGCCCAAGCAGTATAAGCA, *Dio2* (forward primer) TACAAACAGGTTAAACTGGGT-GAAGATGCTC, *Dio2* (reverse primer) GAGCCTCATCAATGTATACCAACAGGAAGTC, *Hprt* (forward primer) GCTGGTGAAAAGGACCTC, *Hprt* (reverse primer) CACAGG ACTAGAACA-CCT, *Prdm16* (forward primer) TCTACATTCCTGAAGACATTC CAATCCCACCA, *Prdm16* (reverse primer) TGTATCCGTCAGCATCTCCCATC CAAAGTC, *Pgc1a* (forward primer) AA-GACAGGTGCCTTCAGTTCACTCTCA G, *Pgc1a* (reverse primer) AGCAGCACACTCTATGT-CACTCCATACAG, *Ucp1* (forward primer) GATGGTGAACCCGACAACTTCCGAAGTG, *Ucp1* (reverse primer) TTCACCTTGGATCTGAAGGCGGACTTTGG, *Il-25* (forward primer) GTGACTGGTGAGCAGTGTCC), *Il-25* (reverse primer) GGTTCCCACGATCAT TGC, *Il-33* (forward primer) GGTGAACATGAGTCCCATCA, *Il-33* (reverser primer) CGTCACCCCTTTGAAGCTC.

**Statistical analysis**. Experiments were repeated at least three times ($n = 4–8$ each) and data are shown as the representative of three independent experiments. A two-tailed student $t$-test for unpaired data was used for comparisons between each group using Prism Software (GraphPad Software Inc.). Error bars represent standard error of the mean. The degree of significance were indicated as: $*p < 0.05$, $**p < 0.01$, $***p < 0.001$.

## Data availability

Sequence data that support the findings of this study (Fig. 7) have been deposited in Genbank with the primary accession code GSE123735. All remaining data will be made available by the corresponding author upon reasonable request.

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

## Acknowledgements
This article was financially supported by National Institutes of Health Public Health Service grants R01 ES025786, R01 ES021801, R21 ES024707, R21 AI109059 (O.A.). B.P.H. is supported by the Swiss National Science Foundation for Early Postdoc.Mobility #181286.

## Author contributions
L.G-T. performed experiments, analyzed the data, designed the figures and wrote the manuscript. I.S. performed experiments and analyzed the data. B.P.H. performed experiments, analyzed the data, designed the figures and wrote the manuscript. E.H. and R.L. helped perform the experiments. H.M., G.L., H.B., H.A., P.S. contributed to data interpretation. F.D.G., A.H.S., and V.K.R provided study guidance. A.L.E. and P.H. provided the GITR agonist and GITR-L-Fc. O.A. supervised the studies and contributed to data interpretation and improvement of the manuscript. All authors contributed to manuscript revision.

## Additional information

**Competing interests:** The authors declare no competing interests.

**Journal Peer Review Information**: *Nature Communications* thanks Carlo Riccardi and the other anonymous reviewers for their contribution to the peer review of this work. Peer reviewer reports are available.

