## [Peer Review File · Nature Communications]

Reviewers' comments:

Reviewer #1 (Remarks to the Author):

Galle-Treger et al. demonstrate that Group 2 innate lymphoid cells (ILC2s) express GTR and that stimulation of IL-33-activated ILC2s with the GTR agonist DTA-1 promotes the secretion of the effector cytokines IL-5, IL-13, GM-CSF, IL-6 and IL-9. Administration of DTA-1 to mice was associated with increased VAT ILC2s and VAT ILC2 effector cytokine production. DTA-1 treatment did not appear to affect weight gain in response to a high fat diet (HFD) or in ob/ob mice but markedly improved glucose and insulin tolerance in vivo in wildtype HFD-fed mice, Rag2^{-/-} HFD-fed mice, and ob/ob mice. DTA-1 treatment was also associated with increased VAT UCP1 expression, increased metabolic rate, and decreased adiposity although this presents a bit of a conundrum in the absence of a weight gain phenotype. ILC2 adoptive transfer experiments into Gitr^{-/-} mice indicate that the GTR/ILC2 axis improves glucose tolerance in an IL-13-dependent manner. While recent work has already established a role for GTR in regulating ILC2 responses in allergic lung inflammation (Nagashima et al, JACI 2018), the role of the GTR/ILC2 axis in metabolic disorders has not yet been reported. This is an interesting, novel manuscript that increases our understanding of how ILC2s control metabolic homeostasis. However, I have a few concerns about the manuscript:

Major:

1. Figure 2: What are ILC2 frequencies and numbers (per gram of fat) in VAT of isotype NCD, DTA-1 NCD, isotype HFD, and DTA-1 HFD mice?
2. Figures 2 and 3: What are IL-5 and IL-13 expression levels in visceral WAT in isotype NCD, DTA-1 NCD, isotype HFD, and DTA-1 HFD mice? What about Rag2^{-/-} HFD treated with or without DTA-1?
3. Figures 2, 3, 5, and 6: Please include VAT weights
4. Figure 3: DTA-1 treatment of Rag2^{-/-} HFD mice appears to be associated with increased metabolic rate, but there is no effect on food intake or activity levels. Together, these outcomes should produce negative caloric balance and lead to weight loss. However body weights are similar in Iso vs DTA-1 in all experiments shown. Why is this?
5. Figure 3i: adiposity is markedly reduced with DTA-1 treatment (from ~45% body fat to ~25% body fat). For 45 gram obese mice, this should translate to ~9 grams of weight loss. However Figs 2b, 3b, and 5b show no effect of DTA-1 treatment on body weight. How do the authors reconcile this?
6. Figure 6: What are ILC2 frequencies and numbers/g VAT for both ILC2 transfer experiments described in Fig 6a (WT ILC2 to Gitr^{-/-} +/- iso or DTA-1) and 6e (WT ILC2 or I15^{-/-} ILC2 or I113^{-/-} ILC2 to Gitr^{-/-} + DTA-1)? Sort-purified ILC2s from VAT have been shown to relocate to VAT in adoptive transfer experiments, but do adoptively transferred VAT ILC2s persist in recipient VAT and retain effector cytokine expression after 14 weeks on a HFD?
7. All of the murine studies rely heavily on the monoclonal agonist DTA-1. Are there any off-target effects of DTA-1? Can the authors rule out an off-target effect of DTA-1 by showing the metabolic phenotype (weight, fasting glucose, GTT) and VAT ILC2 responses in WT HFD vs GTR^{-/-} HFD, each treated with either Isotype or DTA-1?
8. I would like to see a genetic loss-of-function approach that does not involve DTA-1. What is the metabolic phenotype and VAT ILC2 response of Gitr^{-/-} mice on a NCD vs HFD compared to WT controls on a NCD vs HFD?

Minor:

1. The authors might consider citing this paper on the IL-25/ILC2 axis in adipose: Hams et al. J Immunol, 2013 (PMID 24166975).

Reviewer #2 (Remarks to the Author):

The authors demonstrate expression of GITR on ILC2s from both human and mouse. They find that GITR stimulation selectively increases Type 2 cytokine production in activated, but not naïve, ILC2s. Administration of GITR agonist ameliorates the development of insulin resistance in mice on a high-fat diet, despite no change in overall weight gain. This is accompanied by increased energy expenditure, reduced adiposity, and increased expression of thermogenic genes in WAT. The authors demonstrate the requirement of ILC2s for this effect by reproducing these results in T cell-deficient mice, and in GITR-deficient mice receiving adoptive transfer of ILC2s with or without GITR. Beneficial effects are also seen if the agonist is given after onset of obesity. The authors demonstrate that the GITR agonist stimulates the NF κ B signaling pathway and reduces apoptosis in ILC2s. Finally, the authors demonstrate that ILC2s isolated from humans also express GITR and upregulate production of Type 2 cytokines in response to GITR agonist.

This study presents evidence to suggest that GITR on ILC2s may be a useful target for the treatment of metabolic disorders. While the study is sufficiently innovative and interesting to the readership of Nature Communications, the assessment of the metabolic effects of the GITR agonist is inadequate. In order for this study to be suitable for publication in Nature Communications, the following criticisms need to be addressed:

1. Based on the data presented, it is difficult to understand the metabolic effects of the GITR agonist. The authors show that there is no change in weight gain, but adiposity is dramatically reduced (Figure 3). The authors do not describe in their methods how adiposity was determined. The authors should report the results of a DEXA (Echo MRI, etc.), which show both lean and fat mass, and also the weights of individual adipose tissue depots (subcutaneous, visceral, and brown fat depots). The observation of reduced adiposity, but unchanged total mass, means that lean mass must be increased in the agonist-treated mouse. If this is indeed the case, the authors should determine which tissue (skeletal muscle, liver, other?) is the source of this increased lean mass.
2. Furthermore, the simultaneous observations of unchanged weight gain, unchanged caloric intake, and increased energy expenditure in the agonist-treated group are difficult to reconcile, because weight gain (the net accumulation of energy) is completely determined by the combination of caloric intake (energy gained) and energy expenditure (energy lost); if energy expenditure is increased, but caloric intake is unchanged, then theoretically there must be a reduction in weight gain. The authors must be able to explain why their data does not follow this basic rule of energy homeostasis. An alternative interpretation of their data would be that the increase in lean mass, which makes higher contribution to energy expenditure than fat mass, is raising the energy expenditure in the GITR mice. This difference in energy expenditure would thus disappear if the measurements were normalized to total lean mass (which should be larger in GITR treated animals).
3. The authors need to conduct additional experiments to better understand the change in energy expenditure in mice treated with agonist. To do so, they need to characterize brown and beige adipose tissue in these mice. This includes measurement of gross morphology, mass, histology, thermogenic gene expression, and respiration rate (measured by Clarke electrode or analogous instrument) of brown and white adipose tissues. The authors have done some of these experiments, but they need to do more to determine whether the increased energy expenditure is due to altered thermogenic rate in beige and/or brown adipose tissue.
4. The histology images in Figure 3G suggest that the agonist-treated condition has fewer inflammatory cells. This should be confirmed by flow cytometry.

Minor comments:

A. In Figure 1, parts C and D should be consolidated (naïve and activated shown on the same plot).

B. In Figure 2 B, C, D, and E, the plots should show error bars.

C. In many of the bar graphs, the symbols are too large, which makes it not possible to see the error bars in some cases, and more difficult to compare two conditions when the lines are very close to each other.

D. It is considered incorrect to normalize whole mouse VO₂ data to total body weight, because the relationship between VO₂ and body weight depends on body composition, which can be different between experimental groups. It is better to present the VO₂ data without normalizing to body weight.

E. In the Methods, for the RNA-Seq experiment, the time of stimulation with GITR agonist should be indicated.

F. In the Methods, the width of the sections should be "4 μ m" instead of "4 mm".

G. The authors should cite PMID: 25543153 along with ref. 6, when they discuss ILC2s and thermogenesis.

Reviewer #3 (Remarks to the Author):

The paper by Galle-Treger et al contributes with interesting results on the GITR role in ILC2 cells. Recently, preliminary data on the GITR role in regulating ILC2 (Nagashima et al J Allergy Clin Immunol 2018) cells has been described in lung inflammation.

Here authors strongly contribute clearly describing results and giving demonstration that ILC2s co-stimulation by GITR (GITR is mainly a co-stimulatory molecule able to regulate cell activation and survival, as clearly described and discussed in the manuscript) promotes the activity of ILC2 cells and, this is very important, ameliorates diabetes.

Furthermore, in their experiments authors show that effects of GITR triggering are relevant also in animal models lacking the adaptive immunity including the T cell compartment.

This is clearly a significant contribution, new, well presented, very interesting, an advancement in the understanding the GITR function, and suggest a potential new approach to treatment of diabetes.

Furthermore, this contribution is of a broad and general interest even in light of the possibility to develop new therapeutic approaches in different inflammatory and autoimmune diseases.

The manuscript is very well appropriate for publication in Nature Communications.

Minor comments:

Figure 2, may be appropriate to insert (or mention... as an example: "less than 5% of the values") the SD values relatively to the panels b, c, d and e.

Figure 5, also in this figure SD may be inserted in panel b

We would like to thank the reviewers for their critical evaluation of our manuscript. Below we address the comments and concerns of the reviewers in a point-by-point response, referencing the relevant sections of the manuscript. In the marked copy, these modifications have been highlighted in blue.

Reviewer #1 (Remarks to the Author):

Major:

1. Figure 2: What are ILC2 frequencies and numbers (per gram of fat) in VAT of isotype NCD, DTA-1 NCD, isotype HFD, and DTA-1 HFD mice?

We have added the numbers of ILC2s per gram of VAT and the frequencies of ILC2s for these groups. The data is now added to the manuscript as **Supplementary Figure 1b-c**.

2. Figures 2 and 3: What are IL-5 and IL-13 expression levels in visceral WAT in isotype NCD, DTA-1 NCD, isotype HFD, and DTA-1 HFD mice? What about Rag2^{-/-} HFD treated with or without DTA-1?

We have added the number of IL-5⁺ IL-13⁺ ILC2s in VAT from isotype NCD, DTA-1 NCD, isotype HFD, and DTA-1 HFD mice corresponding to Figure 2 in **Supplementary Figure 1d**. The respective percentage of IL-5⁺ and IL-13⁺ ILC2s in VAT from Rag2^{-/-} isotype and DTA-1 HFD corresponding to Figure 3 are shown in **Figure 4c**.

3. Figures 2, 3, 5, and 6: Please include VAT weights

We have added the VAT weights for Figures 2, 3, 5 and 6 as requested and the results are now shown in **Supplementary Figure 1a, Supplementary Figure 3f, Figure 5i and Figure 6i**.

4. Figure 3: DTA-1 treatment of Rag2^{-/-} HFD mice appears to be associated with increased metabolic rate, but there is no effect on food intake or activity levels. Together, these outcomes should produce negative caloric balance and lead to weight loss. However body weights are similar in Iso vs DTA-1 in all experiments shown. Why is this?

We understand the reviewer's point. As mentioned by the reviewer, we observed no weight loss, an increased metabolic rate but did not see any significant effect on food intake or activity levels in response to DTA-1 treatment. Furthermore, we also observed a decrease in the percentage of fat mass (adiposity) after GITR engagement. These results can be explained by a significant increase of the lean mass (now shown in **Supplementary Figure 3e**) in DTA-1 treated mice. Similar results have been observed in WT mice treated with recombinant IL-33 reported by Brestoff *et al.* in Nature in 2015 (PMID: 25533952).

5. Figure 3i: adiposity is markedly reduced with DTA-1 treatment (from ~45% body fat to ~25% body fat). For 45 gram obese mice, this should translate to ~9 grams of weight loss. However Figs 2b, 3b, and 5b show no effect of DTA-1 treatment on body weight. How do the authors reconcile this?

Similarly to our previous answer the fact that the percentage of the lean mass (**Supplementary Figure 3e**) is significantly increased in response to DTA-1 treatment explains the absence of total weight loss in DTA-1 treated mice.

6. Figure 6: What are ILC2 frequencies and numbers/g VAT for both ILC2 transfer experiments described in Fig 6a (WT ILC2 to Gitr^{-/-} +/- iso or DTA-1) and 6e (WT ILC2 or Il5^{-/-} ILC2 or Il13^{-/-} ILC2 to Gitr^{-/-} + DTA-1)? Sort-purified ILC2s from VAT have been shown to relocate to VAT in adoptive transfer experiments, but do adoptively transferred VAT ILC2s persist in recipient VAT and retain effector cytokine expression after 14 weeks on a HFD?

We added the numbers of ILC2/g of VAT for both adoptive transfers in **Figure 6j** as requested. These results suggest that transferred ILC2s are viable and persist in the VAT. Performing an intracellular staining requires fixation and permeabilization which is technically challenging due to the low number of recovered ILC2s. However, we would like to point out that since adoptive transfer of IL-13^{-/-} ILC2s to GITR^{-/-} mice abrogates the protective effect of DTA-1 treatment. This suggests that the transferred WT ILC2s maintain their effector functions in the VAT.

7. All of the murine studies rely heavily on the monoclonal agonist DTA-1. Are there any off-target effects of DTA-1? Can the authors rule out an off-target effect of DTA-1 by showing the metabolic phenotype (weight, fasting glucose, GTT) and VAT ILC2 responses in WT HFD vs GITR^{-/-} HFD, each treated with either Isotype or DTA-1?

We showed in **Figure 2b, c and d** the weight, the fasting blood glucose and the GTT of WT mice fed HFD treated with either Isotype or DTA-1. The VAT ILC2 responses, including numbers and cytokine profile, have now been added in **Supplementary Figure 1b-d**. Furthermore, we also showed in **Figure 6b, c and d** the weight and the fasting blood glucose and the GTT in GITR^{-/-} mice treated either with either Isotype or DTA-1 on HFD. The VAT ILC2 numbers have now been added in **Figure 6j**. Our data show that in absence of GITR expression, we observed no effect of DTA-1 treatment on the metabolic phenotype nor on the

VAT ILC2 numbers. Additionally, in **Figure 6a, b, c and d** the adoptive transfer of WT ILC2s in $GITR^{-/-}$ mice strongly support that the expression of GITR on ILC2s is required to improve the metabolic phenotype. Altogether we believe our results clearly demonstrate the absence of any off-targets effects of DTA-1.

8. I would like to see a genetic loss-of-function approach that does not involve DTA-1. What is the metabolic phenotype and VAT ILC2 response of $Gitr^{-/-}$ mice on a NCD vs HFD compared to WT controls on a NCD vs HFD?

To address the reviewer's comment, the metabolic phenotype and the number of VAT ILC2s of WT mice on NCD and HFD are shown respectively in **Figure 2b-e** and **Supplementary Figure 1b-d**. We have added in **Supplementary Figure 6a-e** the metabolic phenotyping and the VAT ILC2 response of WT vs $GITR^{-/-}$ mice on NCD. In the adoptive transfer experiment of **Figure 6a-d and 6j** we are further showing the metabolic phenotype of $GITR^{-/-}$ on HFD.

We observed that both WT and $GITR^{-/-}$ mice on HFD have increased total weight, fasting blood glucose and are less tolerant to glucose (**Figures 2 and 6**) in response to the diet. Furthermore, they have low numbers of protective VAT ILC2s (**Supplementary Figures 1b and Figure 6j**). However, we observed no difference on the metabolic phenotype and VAT ILC2 response between WT and $GITR^{-/-}$ mice on NCD (**Supplementary Figure 6**). Altogether, we believe our data clearly demonstrate that GITR engagement through its agonist DTA-1 show results consistent with those observed in $GITR^{-/-}$ mice displaying a loss of function.

Minor:

1. The authors might consider citing this paper on the IL-25/ILC2 axis in adipose: Hams et al. J Immunol, 2013 (PMID 24166975).

We have cited this paper as reference 9.

Reviewer #2 (Remarks to the Author):

1. Based on the data presented, it is difficult to understand the metabolic effects of the GITR agonist. The authors show that there is no change in weight gain, but adiposity is dramatically reduced (Figure 3). The authors do not describe in their methods how adiposity was determined. The authors should report the results of a DEXA (Echo MRI, etc.), which show both lean and fat mass, and also the weights of individual adipose tissue depots (subcutaneous, visceral, and brown fat depots). The observation of reduced adiposity, but unchanged total mass, means that lean mass must be increased in the agonist-treated mouse. If this is indeed the case, the authors should determine which tissue (skeletal muscle, liver, other?) is the source of this increased lean mass.

We would like to thank the reviewer for this relevant comment. The adiposity in **Figure 3i** is the fat mass percentage measured by a body composition analyzer (minispec LF90 TD-NMR analyzer) and we accordingly updated our method section. As suggested by the reviewer, we also have added the lean mass percentage in **Supplementary Figure 3e** which shows a significant increase of the lean mass in response to DTA-1 treatment. Since we do not have access to an Echo MRI at USC we were unable to measure the individual adipose tissue depots. However, when the mice were euthanized we measured the weight of the epididymal VAT and the results are now added to the manuscript as **Supplementary Figure 3f**. To determine which tissue is responsible for this increased lean mass, we measured the weights of the liver and spleen (**Supplementary Figure 3g and h respectively**). The results suggest that DTA-1 treatment did not affect the liver and spleen mass. This therefore suggests that tissues other than spleen and liver are responsible for the increased lean mass.

2. Furthermore, the simultaneous observations of unchanged weight gain, unchanged caloric intake, and increased energy expenditure in the agonist-treated group are difficult to reconcile, because weight gain (the net accumulation of energy) is completely determined by the combination of caloric intake (energy gained) and energy expenditure (energy lost); if energy expenditure is increased, but caloric intake is unchanged, then theoretically there must be a reduction in weight gain. The authors must be able to explain why their data does not follow this basic rule of energy homeostasis. An alternative interpretation of their data would be that the increase in lean mass, which makes higher contribution to energy expenditure than fat mass, is raising the energy expenditure in the $GITR$ mice. This difference in energy expenditure would thus disappear if the measurements were normalized to total lean mass (which should be larger in $GITR$ treated animals).

As mentioned previously (please see our response to the questions 4 and 5 of the Reviewer 1), we did in fact observe an increase in the percentage of lean mass (**Supplementary Figure 3e**). Furthermore, as suggested by Reviewer 2, we observed no difference in energy expenditure normalized by lean mass in response to DTA-1. We added these results as **Supplementary Figure 3d**.

3. The authors need to conduct additional experiments to better understand the change in energy expenditure in mice treated with agonist. To do so, they need to characterize brown and beige adipose tissue in these mice. This includes measurement of gross morphology, mass, histology, thermogenic gene expression, and respiration rate (measured by Clarke electrode or analogous instrument) of brown and white adipose tissues. The authors have done some of these experiments, but they need to do more to determine whether the increased energy expenditure is due to altered thermogenic rate in beige and/or brown adipose tissue.

As we currently do not have access to Clarke electrode or an Echo MRI capable of measuring the mass of the different fat pads, we were not able to perform this experiment in a timely manner. However, in response to the reviewer's comment, we added the epididymal VAT mass in all our experiments (**Supplementary Figure 1a, Supplementary Figure 3f, Figure 5i and Figure 6i**) and we already had the histology with UCP1 quantification which is a characteristic gene of brown adipose tissue. Furthermore, in **Supplementary Figure 4** we have the expression by RT-qPCR of five thermogenic genes in the VAT. Several groups have demonstrated that these five genes are molecular markers of brown adipose tissue, in particular, Dio2, Prdm16 and Pgc1a expressions are also positively associated with a higher metabolic rate (PMID: 21698184; PMID: 24439384; PMID: 22237023). As these genes are significantly increased in response to DTA-1 treatment, we believe our results demonstrate that the engagement of GTR induces a higher metabolic rate. Additionally, Brestoff *et al.* (PMID: 25533952) clearly demonstrated that activation of ILC2s induces beiging of the adipose tissue and increased the metabolic rate of the VAT. We now cited and discussed the effect of beiging and ILC2s in the manuscript.

4. The histology images in Figure 3G suggest that the agonist-treated condition has fewer inflammatory cells. This should be confirmed by flow cytometry.

We agree with the reviewer and to address this point, we quantified by flow cytometry the number of CD45⁺ cells in the stromal vascular fraction (SVF) of the VAT (**Figure 4a**) and the total number of macrophages (CD45⁺ CD11b^{hi} F4/80^{hi} cells) per gram of VAT (**Figure 4d**). Our results clearly suggest that in response to GTR engagement the mice have less leukocytes and in particular less macrophages in the VAT.

Minor comments:

A. In Figure 1, parts C and D should be consolidated (naïve and activated shown on the same plot).

We have adapted the **Figure 1c** based on the reviewer's comment.

B. In Figure 2 B, C, D, and E, the plots should show error bars.

We added error bars to the **Figures 2 B, C, D, and E**.

C. In many of the bar graphs, the symbols are too large, which makes it not possible to see the error bars in some cases, and more difficult to compare two conditions when the lines are very close to each other.

We edited the figures.

D. It is considered incorrect to normalize whole mouse VO₂ data to total body weight, because the relationship between VO₂ and body weight depends on body composition, which can be different between experimental groups. It is better to present the VO₂ data without normalizing to body weight.

We have adapted **Figure 3j** based on the reviewer's comment and now show the VO₂ without normalization.

E. In the Methods, for the RNA-Seq experiment, the time of stimulation with GTR agonist should be indicated.

We have added the time of stimulation.

F. In the Methods, the width of the sections should be "4 μm" instead of "4 mm".

We have corrected this mistake.

G. The authors should cite PMID: 25543153 along with ref. 6, when they discuss ILC2s and thermogenesis.

We have cited this manuscript as reference 8.

Reviewer #3 (Remarks to the Author):

Minor comments:

Figure 2, may be appropriate to insert (or mention... as an example: "less than 5% of the values") the SD values relatively to the panels b, c, d and e.

We apologize for our mistake. The error bars from **Fig. 2b-e** were missing and are now added to the above-mentioned figures.

Figure 5, also in this figure SD may be inserted in panel b

We have now adapted the error bars of **Figure 5b** as requested.

REVIEWERS' COMMENTS:

Reviewer #1 (Remarks to the Author):

The authors have addressed my concerns. I have no further comments.

Reviewer #2 (Remarks to the Author):

The authors have sufficiently addressed my concerns and the manuscript is now suitable for publication.

Reviewer #3 (Remarks to the Author):

The manuscript has been revised on the basis of referee's indication and data from new experiments are now in the Figures as well new comments in the text The manuscript has been clearly improved.

REVIEWERS' COMMENTS:

Reviewer #1 (Remarks to the Author):

The authors have addressed my concerns. I have no further comments.

Reviewer #2 (Remarks to the Author):

The authors have sufficiently addressed my concerns and the manuscript is now suitable for publication.

Reviewer #3 (Remarks to the Author):

The manuscript has been revised on the basis of referee's indication and data from new experiments are now in the

Figures as well new comments in the text The manuscript has been clearly improved.

We thank the reviewers for their positive feedbacks.